# Lightning-Fast Image Inversion and Editing for Text-to-Image Diffusion Models

**Dvir Samuel**[1,3] *  **Barak Meiri**[1,2]  **Haggai Maron**[4,5]  **Yoad Tewel**[2,5]  **Nir Darshan**[1]

**Shai Avidan**[2]  **Gal Chechik**[3,5]  **Rami Ben-Ari**[1]

[1]OriginAI, [2]Tel-Aviv University, [3]Bar-Ilan University, [4]Technion, [5]NVIDIA Research

Israel

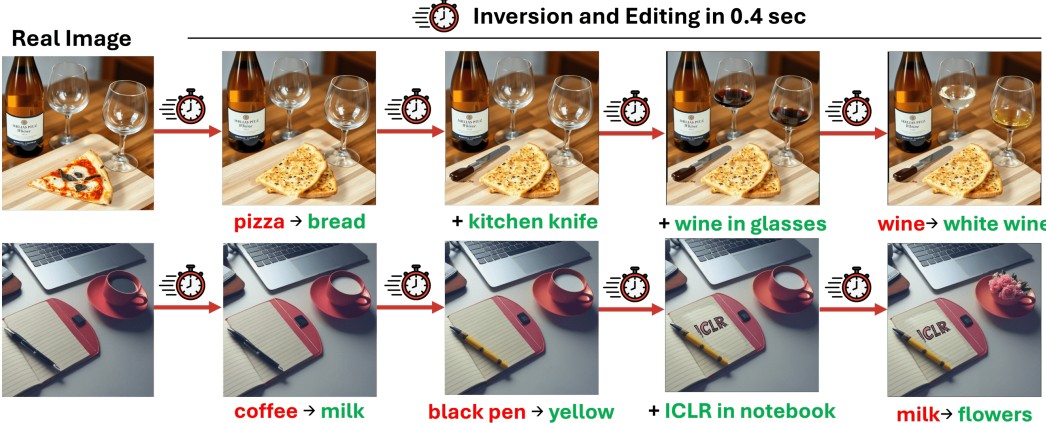

Figure 1: Consecutive real image inversions and editing using our GNRI with Flux.1-schnell (Black-Forest, 2024) (0.4 sec on an A100 GPU).

## Abstract

Diffusion inversion is the problem of taking an image and a text prompt that describes it and finding a noise latent that would generate the exact same image. Most current deterministic inversion techniques operate by approximately solving an implicit equation and may converge slowly or yield poor reconstructed images. We formulate the problem by finding the roots of an implicit equation and devlop a method to solve it efficiently. Our solution is based on Newton-Raphson (NR), a well-known technique in numerical analysis. We show that a vanilla application of NR is computationally infeasible while naively transforming it to a computationally tractable alternative tends to converge to out-of-distribution solutions, resulting in poor reconstruction and editing. We therefore derive an efficient guided formulation that fastly converges and provides high-quality reconstructions and editing. We showcase our method on real image editing with three popular open-sourced diffusion models: Stable Diffusion, SDXL-Turbo, and Flux with different deterministic schedulers. Our solution, **Guided Newton-Raphson Inversion**, inverts an image within 0.4 sec (on an A100 GPU) for few-step models (SDXL-Turbo and Flux.1), opening the door for interactive image editing. We further show improved results in image interpolation and generation of rare objects.

## 1 Introduction

Text-to-image diffusion models (Rombach et al., 2022; Saharia et al., 2022; Ramesh et al., 2022; Balaji et al., 2022) can generate diverse and high-fidelity images based on user-provided text prompts. These models are further used in several important tasks that require *inversion*, namely, discovering an initial noise (seed) that, when subjected to a backward (denoising) diffusion process along

---

*Correspondence to: Dvir Samuel dvirsamuel@gmail.com

with the prompt, generates the input image. Inversion is used in various tasks including image editing (Hertz et al., 2022), personalization (Gal et al., 2023b;a), seed noise interpolation for semantic augmentation (Samuel et al., 2023) and generating rare concepts (Samuel et al., 2024).

As inversion became a critical building block in various tasks, several inversion methods have been suggested. Denoising Diffusion Implicit Models (DDIM) (Song et al., 2021) introduced a deterministic and fast sampling technique for image generation with diffusion models. DDIM *inversion* transforms an image back into a latent noise representation by approximating the inversion equation. Although this approximation makes it very fast, it also introduces an approximation error (as explained in section 3), causing noticeable distortion artifacts in the reconstructed images. This is particularly noticeable in few-step diffusion models (Luo et al., 2023; Sauer et al., 2023; Esser et al., 2024), with large gaps between the diffusion time-steps, and where inference is achieved with only $2 - 4$ steps. Several efforts have been made to address inconsistencies in DDIM Inversion, which result in poor reconstruction quality (Mokady et al., 2023; Wallace et al., 2023; Pan et al., 2023). For example, (Pan et al., 2023; Garibi et al., 2024) used fixed-point iterations to solve the DDIM-inversion equation. Another approach in (Hong et al., 2024) tackled the minimization of residual error in DDIM-inversion via gradient descent. Although these methods demonstrate improvements over previous approaches, their editing quality and computational speed are still limited.

In this paper, we frame the deterministic diffusion inversion problem as finding the roots of an implicit function. We propose a solution based on the Newton-Raphson (NR) numerical scheme (Burden et al., 2015) - a very fast and well-tested optimization method for root finding. We further define a scalar NR scheme for this problem that can be computed efficiently. However, NR has a main disadvantage in our context. When applied to our highly non-convex function, it tends to find roots that drift outside the distribution of latents that the diffusion model was trained on.To mitigate this, we introduce a guidance term that leverages prior knowledge of the likely solution locations, steering the NR iterations toward in-distribution roots. This is done by adding this prior knowledge on the noise distribution at each diffusion step. Ultimately, our approach enables rapid inversion while maintaining state-of-the-art reconstruction and editing quality.

We name our approach GNRI, for **Guided Newton Raphson Inversion**. GNRI converges in just a few iteration steps at each diffusion time step and achieves high-quality image inversions (in terms of reconstruction accuracy). In practice, 1-2 iterations are sufficient for convergence that yields significantly more accurate results than other inversion methods. Importantly, GNRI requires no model training, fine-tuning, prompt optimization, or additional parameters, and is compatible with all pre-trained diffusion and flow-matching models with deterministic schedulers. We demonstrate its effectiveness for inverting different deterministic schedulers used with latent diffusion model (Stable Diffusion) (Rombach et al., 2022), few-step latent diffusion model (SDXL-turbo) (Sauer et al., 2023) and few-step flow matching model (Flux.1) (Black-Forest, 2024). Figure 1 demonstrates the quality and speed of GNRI for iterative editing using few-step diffusion models. Using such models models that require 4 denoising steps, our approach can edit real images within 0.4 seconds. This allows users to edit images *on the fly*, using text-to-image models.

We conduct a comprehensive evaluation of GNRI. First, we directly assess the quality of inversions found with GNRI by measuring reconstruction errors compared to deterministic inversion approaches. Our method suppresses all methods with $\times 2$ to $\times 40$ speedup gain. We then demonstrate the benefit of GNRI in two downstream tasks (1) In *Image editing*, GNRI smoothly changes fine details in the image in a consistent and coherent way, whereas previous methods struggle to do so. (2) In *Seed Interpolation* and *Rare concept generation* (Samuel et al., 2023) that require diffusion inversion. In both of these tasks, GNRI yields more accurate seeds, resulting in superior generated images, both qualitatively and quantitatively.

## 2 RELATED WORK

Text-to-image diffusion models (Rombach et al., 2022; Saharia et al., 2022; Ramesh et al., 2022; Balaji et al., 2022; Esser et al., 2024) translate random samples (seeds) from a high-dimensional space, guided by a user-supplied text prompt, into corresponding images. DDIM (Song et al., 2021) is a widely used deterministic scheduler, that demonstrates the inversion of an image to its latent noise seed. When applied to the inversion of text-guided diffusion models, DDIM inversion suffers from low reconstruction accuracy that is reflected in further tasks, particularly when applied to few-

step diffusion models that require only 3-4 denoising steps. This happens because DDIM inversion relies on a linear approximation, causing the propagation of errors that result in inaccurate image reconstruction. Recent studies (Mokady et al., 2023; Wallace et al., 2023; Pan et al., 2023; Hong et al., 2024) address this limitation. Null-text inversion (Mokady et al., 2023) optimizes the embedding vector of an empty string. This ensures that the diffusion process calculated using DDIM inversion, aligns with the reverse diffusion process. (Miyake et al., 2023) replace the null-text embedding with a prompt embedding instead. This enhances convergence time and reconstruction quality but results in inferior image editing performance. In both (Mokady et al., 2023) and (Miyake et al., 2023), the optimized embedding must be stored, resulting in nearly 3 million additional parameters for each image (using 50 denoising steps of StableDiffusion (Rombach et al., 2022)).

EDICT (Wallace et al., 2023) introduced invertible neural network layers, specifically Affine Coupling Layers, to calculate both backward and forward diffusion paths. While effective, it comes at the cost of prolonging inversion time. BDIA (Zhang et al., 2023) introduced a novel approximation tailored for EDICT, enhancing its computational efficiency while maintaining the accuracy in diffusion inversion. However, it still demands considerably more time, approximately ten times longer than DDIM Inversion. TurboEdit (Wu et al., 2024) introduced an encoder-based iterative inversion technique for few-step diffusion models, enabling precise image inversion and disentangled image editing. DirectInv (Ju et al., 2023) separates the source and target diffusion branches for improved text-guided image editing, leading to better content preservation and edit fidelity. AIDI (Pan et al., 2023) and ReNoise (Garibi et al., 2024) used a fixed-point iteration technique at each inversion step to find a better solution for the implicit function posed by DDIM equations and thereby achieving improved accuracy. ExactDPM (Hong et al., 2024), similar to (Pan et al., 2023; Garibi et al., 2024) proposed gradient-based methods for finding effective solutions for the implicit function.

Alternative methods proposed in (Huberman et al., 2023; Brack et al., 2024; Deutch et al., 2024) use a stochastic DDPM inversion instead of a deterministic one. Although it can recover the exact image during reconstruction, the stochastic nature of these approaches usually causes the method to struggle with reconstructing fine details during editing. These methods require many steps for high-quality editing, making the process time-consuming. Additionally, they demand a larger memory footprint, as they need to store $T + 1$ latents for each inverted image, restricting their use only to reconstruction and editing tasks. We also compare our approach with these methods in terms of editing quality and demonstrate that our method outperforms them all.

## 3 PRELIMINARIES

We first establish the fundamentals of Denoising Diffusion Implicit Models (DDIMs). In this model, a *backward pass* (denoising) is the process that generates an image from a seed noise. A *forward pass* is the process of adding noise gradually to an image until it becomes pure Gaussian noise. *Inversion* (Song et al., 2021) is similar to the forward pass but the goal is to end with a specific Gaussian noise that would generate the image if denoised.

**Forward Pass in Diffusion Models.** Diffusion models (Rombach et al., 2022) learn to generate images through a systematic process of iteratively adding Gaussian noise to a latent data sample until the data distribution is mostly noise. The data distribution is subsequently gradually restored through a reverse diffusion process initiated with a random sample (noise seed) from a Gaussian distribution. In more detail, the process of mapping a (latent) image to noise is a Markov chain that starts with $z_0$, and gradually adds noise to obtain latent variables $z_1, z_2, \ldots, z_T$, following a distribution $q(z_1, z_2, \ldots, z_T | z_0) = \Pi_{t=1}^{T} q(z_t | z_{t-1})$, where $\forall t : z_t \in \mathbb{R}^d$ with $d$ denoting the dimension of the space. Each step in this process is a Gaussian transition, that is, $q$ follows a Gaussian distribution

$$q(z_t | z_{t-1}) := \mathcal{N}(z_t; \mu_t = \gamma_t z_{t-1}, \Sigma_t = \beta_t I), \tag{1}$$

parameterized by a schedule $(\beta_0, \gamma_0), \ldots, (\beta_T, \gamma_T) \in (0, 1)^2$. As discussed below, in DDIM, $\gamma_t = \sqrt{1 - \beta_t}$ and in Euler scheduling $\beta_t = \gamma_t = 1 \, \forall t$.

**Deterministic schedule diffusion models.** It has been shown that one can "sample" in a deterministic way from the diffusion model, and this accelerates significantly the denoising process. Several deterministic schedulers have been proposed (Song et al., 2021; Lu et al., 2022; Karras et al., 2022), we describe here two popular ones, DDIM and Euler schedulers.

*Denoising Diffusion Implicit Models (DDIM).* Sampling from diffusion models can be viewed as solving the corresponding diffusion Ordinary Differential Equations (ODEs) (Lu et al., 2022). DDIM (Song et al., 2021), a popular deterministic scheduler, proposed to denoise a latent by

$$z_{t-1} = \sqrt{\frac{\alpha_{t-1}}{\alpha_t}} z_t - \sqrt{\alpha_{t-1}} \cdot \Delta\psi(\alpha_t) \cdot \epsilon_\theta(z_t, t, p), \tag{2}$$

where $\alpha_t = 1 - \beta_t$, $\psi(\alpha) = \sqrt{\frac{1}{\alpha} - 1}$, and $\Delta\psi(\alpha_t) = \psi(\alpha_t) - \psi(\alpha_{t-1})$ and $\epsilon_\theta(z_t, t, p)$ is the output of a network that was trained to predict the noise to be removed.

*Euler schedulers.* Euler schedulers follow a similar deterministic update rule

$$z_{t-1} = z_t + (\sigma_{t-1} - \sigma_t) v_\theta(z_t, t, p), \tag{3}$$

here $\sigma_t, \sigma_t - 1$ are scheduling parameters and $v_\theta$ is the output of the network that was trained to predict the velocity.

**Diffusion inversion.** We now focus on inversion in the latent representation. We describe our approach first for DDIM, and extension to other schedulers is discussed below. Given an image representation $z_0$ and its corresponding text prompt $p$, we seek a noise seed $z_T$ that, when denoised, reconstructs the latent $z_0$. Several approaches were proposed for this task. DDIM inversion rewrites Eq. (2) as:

$$z_t = f(z_t)$$

$$f(z_t) \quad := \quad \sqrt{\frac{\alpha_t}{\alpha_{t-1}}} z_{t-1} + \sqrt{\alpha_t} \cdot \Delta\psi(\alpha_t) \cdot \epsilon_\theta(z_t, t, p). \tag{4}$$

DDIM inversion approximates this implicit equation in $z_t$ by replacing $z_t$ with $z_{t-1}$

$$f(z_t) \approx \sqrt{\frac{\alpha_t}{\alpha_{t-1}}} z_{t-1} + \sqrt{\alpha_t} \cdot \Delta\psi(\alpha_t) \cdot \epsilon_\theta(z_{t-1}, t, p). \tag{5}$$

The quality of the approximation depends on the difference $z_t - z_{t-1}$ (a smaller difference would yield a small error) and on the sensitivity of $\epsilon_\theta$ to that $z_t$. See (Dhariwal & Nichol, 2021; Song et al., 2021) for details. By applying Eq. (5) repeatedly for every denoising step $t$, one can invert an image latent $z_0$ to a latent $z_T$ in the seed space.

DDIM inversion is fast, but the approximation of Eq. (5) inherently introduces errors at each time step. As these errors accumulate, they cause the whole diffusion process to become inconsistent in the forward and the backward processes, leading to poor image reconstruction and editing (Mokady et al., 2023; Wallace et al., 2023; Pan et al., 2023). This is particularly noticeable in few-step and consistency models with a small number of denoising steps (typically 2-4 steps), where there's a significant gap between $z_t$ and $z_{t-1}$ Garibi et al. (2024).

This inversion technique can also be applied to other deterministic schedulers. For instance, for Euler scheduler, one defines $z_t = f(z_t)$, $f(z_t) := z_{t-1} + (\sigma_t - \sigma_{t-1}) v_\theta(z_t, t, p)$, and this is approximated by $z_{t-1} + (\sigma_t - \sigma_{t-1}) v_\theta(z_{t-1}, t, p)$.

**Iterative inversion and optimization methods.** Several papers proposed to improve the approximation using iterative methods. AIDI (Pan et al., 2023) and ReNoise (Garibi et al., 2024) proposed to directly solve Eq. (4) using fixed-point iterations (Burden, 1985), a widely-used method in numerical analysis for solving implicit functions. In a related way, (Hong et al., 2024) solves a more precise inversion equation, obtained by employing higher-order terms, using gradient descent.

**Newton-Raphson Root finding.** The Newton-Raphson method is a widely used numerical technique for finding roots of a real-valued function $F$ (Burden et al., 2015). It is particularly effective for solving equations of the form $F(z) = 0$, when $F : \mathbb{R}^D \to \mathbb{R}^D$ for an arbitrary dimension $D$. It provides fast convergence, typically quadratic, by requiring the evaluation of the function and the inversion of its Jacobian matrix. It has also been shown that when initialized near a local extremum, NR may converge to that point (oscillating around it) (Kaw et al., 2003). The NR scheme in general form, is given by

$$z_t^{k+1} = z_t^k - J(z_t^k)^{-1} F(z_t^k), \tag{6}$$

where $J(z_t^k)^{-1}$ presents the inverse of a Jacobian matrix $J \in \mathbb{R}^{D \times D}$ for $F$ (all derivatives are w.r.t to $z$), and $k$ stands for the iteration number. The iteration starts with an initial guess, $z = z^0$.

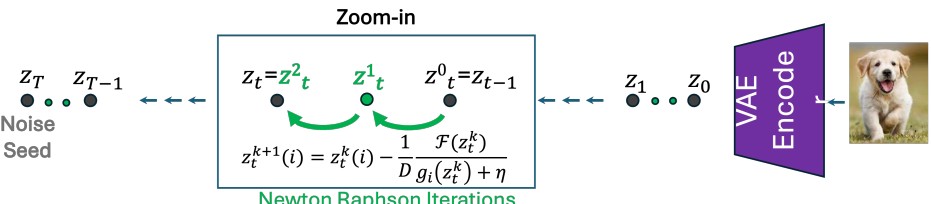

Figure 2: **Newton-Raphson Inversion** iterates over an implicit function Eq. 4 using Eq. 10 scheme, at every time-step in the inversion path. Initialized with $z_t^0 = z_{t-1}$ it converges within $\approx 2$ iterations, to $z_t$. Each box denotes one inversion step; black circles correspond to intermediate latents in the denoising process; green circles correspond to intermediate Newton-Raphson iterations.

## 4 OUR METHOD: GUIDED NEWTON RAPHSON INVERSION

Existing inversion methods are either fast-but-inaccurate, like DDIM inversion, or more precise-but-slow, like (Pan et al., 2023) and (Hong et al., 2024). This paper presents a method that achieves a balance of speed and improved precision compared to these existing approaches.

To achieve these features we introduce two key ideas. First, we frame inversion as a scalar root-finding problem that can be solved efficiently, and use the well-known *Newton-Raphson* root-finding method. Second, we make the observation that an inversion step is equivalent to finding a specific noise vector, and we *use a strong prior* obtained from the distribution of that noise. This prior during the iterative solution guides Newton-Raphson towards finding a root that adheres to the correct distribution of latents. See Figure 2 for illustration.

### 4.1 FRAMING INVERSION AS EFFICIENT ROOT-FINDING

Diffusion inversion can be done by finding the fixed point of the function $f(z)$ in Eq. 4. This is equivalent to finding the zero-crossings or roots, $z_t$ of the residual function $r : \mathbb{R}^D \to \mathbb{R}^D$ defined as $r(z_t) := z_t - f(z_t)$. One could now apply *Newton-Raphson* (NR) to find a root. However, in our case, $z_t$ is a high dimensional latent($D \approx 16K$ in StableDiffusion (Rombach et al., 2022)), making it computationally infeasible to compute the Jacobian (Eq. 6), explicitly and invert it. To address this computational limitation, we apply a norm over $r(z_t)$, $\hat{r} : \mathbb{R}^d \to \mathbb{R}_0^+$ yielding a multi-variable, *scalar* function, which has the same set of roots:

$$\hat{r}(z_t) \quad := \quad ||z_t - f(z_t)||_1 \quad , \tag{7}$$

and seek the roots $\hat{r}(z_t) = 0$ where $||\cdot||$ denotes a $L_1$ norm which simply sum over all absolute values of $\hat{r}$. Applying the norm on the residual has be done in previous work (Hong et al., 2024; Garibi et al., 2024), however here it is used to reduce the Jacobian to a vector and that can be computed quickly (see derivation in Appendix A). We call this method *Newton-Raphson Inversion*, NRI.

This method is efficient and converges quickly in practice. However, NRI in our case has a key limitation: Eq. (4), and thus $\hat{r}$, can have multiple solutions or local minima. The accelerated convergence of NR due to linear "extrapolation", can cause it to "jump" to solutions outside the latent variable distribution for the diffusion model. Indeed, in practice, we find that solving Eq. (7) for $\hat{r}(z_t) = 0$ often converges quickly to a solution that results with poor image reconstruction. We quantify and illustrate this phenomenon in section 5.1. Next, we describe a remedy to this problem.

### 4.2 GUIDED NEWTON RAPHSON INVERSION

The NR procedure discussed above may find roots that are far from the modes of the latent distribution. How can we guide NR to converge to useful roots among all possible ones?

Note that determining a $z_t$ given $z_{t-1}$, is equivalent to finding which noise term is added during the forward pass of the diffusion process. Luckily, the distribution of that noise is known by the design of the diffusion process and we can use it as prior information. More precisely, since each step in the diffusion process follows a Gaussian distribution $q(z_t|z_{t-1})$ (Eq. 1), we define a positive function

that obtains a zero value at the maximum likelihood of the $q$ distribution by taking the negative log of the exponent part of the distribution (without the normalizing factor). This can be simplified to

$$G(z_t) = \begin{cases} \frac{1}{\beta_t}||z_t - \mu_t|| & \text{DDIM,} \\ ||z_t - \mu_t|| & \text{Euler.} \end{cases} \tag{8}$$

where $\mu_t$, $\beta_t$ were defined in Section 3, see Eq (1). To guide NR to select an "in-distribution" solution, we define a new objective $\mathcal{F} : \mathbb{R}^D \to \mathbb{R}_0^+$:

$$\mathcal{F}(z_t) := ||z_t - f(z_t)||_1 + \lambda G(z_t). \tag{9}$$

Here, $\lambda > 0$ is a hyperparameter weighting factor for the guidance term. See Appendix E for ablation of $\lambda$ values. Note that this guidance term incorporates prior knowledge about the probable locations of the solutions in $z$ space.

We then seek to drive $\mathcal{F}(z_t)$ near zero. Applying the Newton Raphson iteration scheme for Eq. (9) then emerges as (see derivation in Appendix A):

$$z_t^0 = z_{t-1}$$
$$z_t^{k+1}(i) = z_t^k(i) - \frac{1}{D} \frac{\mathcal{F}(z_t^k)}{g_i(z_t^k) + \eta}. \tag{10}$$

Here $g_i := \frac{\partial \mathcal{F}(z_t)}{\partial z_t(i)}$ is the partial derivative of $\mathcal{F}$ with respect to the variable (descriptor) $z_t(i)$. The index $i \in [0, 1, ..., D]$ indicates $z$'s components and $k$ stands for the iteration number, while $\eta$ is a small constant added for numerical stability. The function $g$ can be computed efficiently using automatic differentiation engines. We initialize the process with $z$ from the previous diffusion time-step $z_{t-1}$. We call our approach **GNRI** for **G**uided **N**ewton **R**aphson **I**nversion. For discussion on applying NR on our objective function and the guidance term see Appendix B.

## 5 EXPERIMENTS

Here, we first provide **motivation and analysis** for the guidance term presented in Section 4.2. Then, we show evaluation results of GNRI on three main tasks: (1) **Image inversion and reconstruction:** Here, We assess the inversion fidelity by evaluating the quality of the reconstructed images. (2) **Image Editing:** We demonstrate the efficacy of our inversion scheme in image editing, highlighting its ability to facilitate real-time editing. (3) **Rare Concept Generation:** We illustrate how our method can be applied to improve the generation of rare concepts.

**Compared Methods:** We compared our approach with the following methods. Standard **DDIM Inversion** (Song et al., 2021), **SDEdit** Meng et al. (2022), **Null-text** (Mokady et al., 2023) and **EDICT** (Wallace et al., 2023). Two fixed-point based methods: **AIDI** (Pan et al., 2023) and **ReNoise** (Garibi et al., 2024) with further comparison to Anderson Acceleration Pan et al. (2023) shown in Appendix I. Gradient-based method **ExactDPM** (Hong et al., 2024). **Wu2024Turbo** (Wu et al., 2024) and two DDPM inversion methods: **EditFriendly** Huberman et al. (2023) and **Gal2024Turbo** (Deutch et al., 2024). In all experiments, we used code and parameters provided by the respective authors.

**Implementation details:** To demonstrate the versatility of our approach, we conducted experiments on latent diffusion model SD2.1 Rombach et al. (2022) and two few-step models: SDXL-turbo (Sauer et al., 2023) and the newly introduced flow matching Flux.1-schnell model (Black-Forest, 2024). Sampling steps were set to 50 for SD2.1 and 4 for SDXL-turbo and Flux.1. All methods ran until convergence. All methods were tested on a single A100 GPU for a fair comparison. PyTorch's built-in gradient calculation was used for computing derivatives of Eq. (9). Editing for all iterative methods was done using Prompt-to-Prompt (Hertz et al., 2022).

### 5.1 GUIDANCE TERM MOTIVATION AND COMPARISON TO OTHER NUMERICAL SCHEMES

To illustrate the motivation behind our guidance term, we compare the reconstruction quality of non-guided NRI a "vanilla" NR inversion method with two iterative numerical schemes: Fixed-point Iteration (FPI) ((Pan et al., 2023) and Gradient Descent (GD) (Hong et al. (2024). We test the

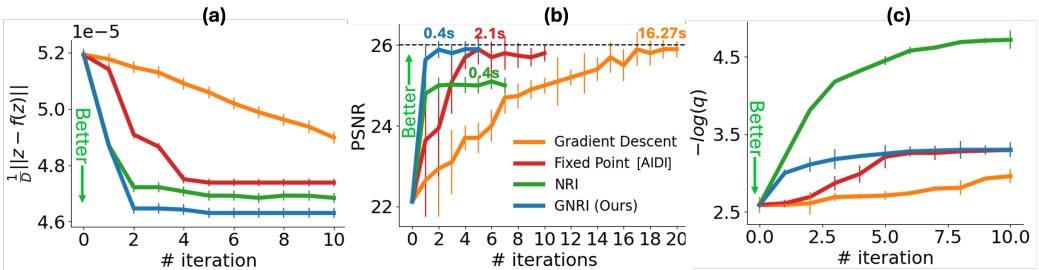

Figure 3: The effect of GNRI guiding term on NR inversion, and comparison to other iterative inversion methods. All results are averages computed with SDXL-Turbo applied to 5,000 COCO images. **(a) Average residuals throughout optimization.** NR-based methods are the fastest to converge. Gradient-descent was run with the largest learning rate that was stable but still is slowest. **(b) Reconstruction quality (PSNR).** Adding guidance (blue) to NRI (green) significantly improves the quality of the converged solution. **(c) Likelihood of inferred noise.** Without the guiding term, NRI (green) finds solutions that are substantially different from those found by other methods, which explains the low reconstruction quality.

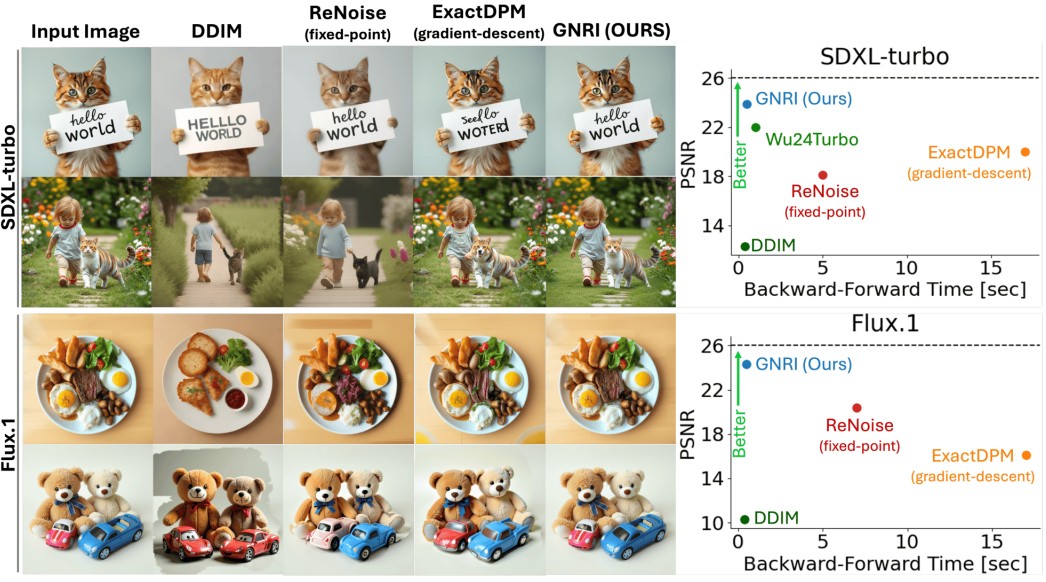

Figure 4: **(Left) Reconstruction qualitative results:** Comparing image inversion-reconstruction performance. While all baseline methods struggle to preserve the original image, GNRI successfully excels in accurately reconstructing it. **(Right) Inversion Results:** Mean reconstruction quality (y-axis, PSNR) and runtime (x-axis, seconds) on the COCO2017 validation set. Our method achieves high PSNR while reducing inversion-reconstruction time by a factor of $\times 2$ (compared to DDIM) and up to $\times 40$ (compared to ExactDPM) on SDXL-turbo and $\times 10$ to $\times 40$ on Flux.1, compared to other approaches.

reconstruction of 5,000 image-caption pairs from COCO Lin et al. (2014), presenting three metrics in Fig. 3. Fig 3a shows that FPI, GD and non-guided NRI, all converge to low residual values but the reconstruction values for NRI (green line) in Fig. 3b are notably lower. This suggests that solutions reached by NRI are in areas of the latent space that yield poor results. Comparing the blue curve in Fig 3 to the green curve highlights the significant impact of adding a guidance term. It accelerates convergence (Fig. 3a), improves reconstruction (PSNR) (Fig 3b) and reaches a better likelihood (Fig. 3c).

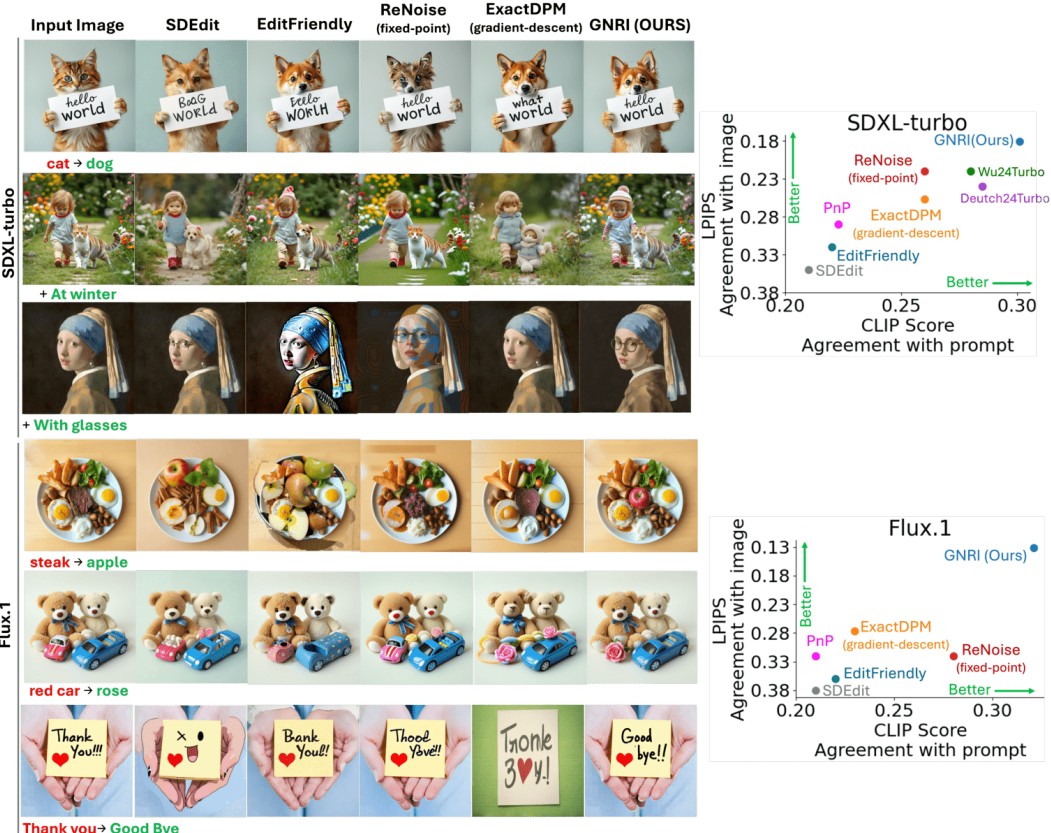

Figure 5: (**Left**) **Qualitative results of image editing**. GNRI edits images more naturally while preserving the structure of the original image. All baselines were executed until they reached convergence. (**Right**) **Evaluation of editing performance:** GNRI achieves superior CLIP and LPIPS scores, indicating better compliance with text prompts and higher structure preservation.

## 5.2 IMAGE RECONSTRUCTION

To evaluate the fidelity of our approach, we measure the PSNR of images reconstructed from seed inversions. Specifically, we used the entire set of 5000 images from the MS-COCO-2017 validation dataset (Lin et al., 2014), along with their corresponding captions. For each image-caption pair, we first found the inverted latent and then used it to reconstruct the image using the same caption. Given that the COCO dataset provides multiple captions for each image, we used the first listed caption as a conditioning prompt. Figure 4 (left) demonstrates a qualitative comparison of reconstructed images. Figure 4 (right) illustrates the PSNR of reconstructed images versus inversion time, highlighting the performance of our method compared to SoTA inversion techniques. The dashed black line represents the upper bound set by the Stable Diffusion VAE, showing that our method achieves the highest PSNR among SoTA methods and approaches the VAE upper bound, while also being the fastest. Additional qualitative comparisons for SD2.1 and further analysis are in Appendix H.

## 5.3 REAL-TIME IMAGE EDITING

Current state-of-the-art methods for editing images start by inverting a real image into latent space and then operating on the latent. As a result, the quality of editing depends strongly on the quality of inversion (Mokady et al., 2023).

We now evaluate the effect of inverting images with GNRI, compared to SoTA inversion baselines. We show that GNRI outperforms all baselines in editing performance while requiring the shortest time. This opens the door for **real-time editing capabilities**. We present below both qualitative and quantitative results. For Neural Function Evaluations (NFE) comparison see Appendix H.

Table 1: Image editing comparison using descriptive text in PIE-Bench (Ju et al., 2023) dataset with SDXL-turbo. The efficiency is measured in a single A100 GPU. Our method achieves the best background preservation and clip similarity while being significantly faster than other methods.

| PIE-Bench | Background Preservation | | | | CLIP Similarity | | Efficiency | |
|---|---|---|---|---|---|---|---|---|
| | PSNR $\uparrow$ | LPIPS $10^3 \downarrow$ | MSE $10^4 \downarrow$ | SSIM $10^2 \uparrow$ | Whole $\uparrow$ | Edited $\uparrow$ | Inverse (s)$\downarrow$ | Forward (s)$\downarrow$ |
| DDIM | 18.59 | 177.96 | 184.69 | 66.86 | 23.62 | 21.20 | **0.454** | **0.445** |
| ReNoise (Garibi et al., 2024) | 27.11 | 49.25 | 31.23 | 72.3 | 23.98 | 21.26 | 5.41 | 0.445 |
| ExactDPM (Hong et al., 2024) | 24.54 | 59.88 | 36.49 | 69.18 | 23.77 | 21.23 | 15.11 | 0.445 |
| EditFriendly (Huberman et al., 2023) | 24.55 | 91.88 | 95.58 | 81.57 | 23.97 | 21.03 | 32.3 | 23.1 |
| Direct Inv (Ju et al., 2023) | 27.22 | 54.55 | 62.86 | 84.76 | 25.02 | 22.10 | 4.3 | 50 |
| TurboEdit (Wu et al., 2024) | 29.52 | 44.74 | 26.08 | 91.59 | 25.05 | 22.34 | 0.981 | 0.569 |
| GNRI **(Ours)** | **30.22** | **40.15** | **20.10** | **98.66** | **26.15** | **23.26** | 0.521 | 0.445 |

| | Interpolation | | Centroid | |
|---|---|---|---|---|
| | ACC $\uparrow$ | FID $\downarrow$ | ACC $\uparrow$ | FID $\downarrow$ |
| DDIM Inversion | 51.59 | 6.78 | 67.24 | 5.48 |
| ExactDPM (Hong et al., 2024) | 52.01 | 6.13 | 68.14 | 5.32 |
| ReNoise (Garibi et al., 2024) | 52.51 | 6.0 | 69.14 | 5.22 |
| GNRI (**ours**) | **54.98** | **5.91** | **70.18** | **4.59** |

Table 2: **Image interpolation and centroid finding.** In interpolation, two images $x^1, x^2$ are inverted to generate images between their seeds $z_T^1, z_T^2$. In centroid-finding, a set of images is inverted to find their centroid. Best performance is achieved with our GNRI.

**Qualitative results.** Figure 5(left) gives a qualitative comparison between SoTA approaches and GNRI in the context of real image editing. GNRI excels in accurately editing with target prompts producing images with both high fidelity to the input image and adherence to the target prompt. The examples illustrate how alternative approaches may struggle to retain the original structure or tend to overlook crucial components specified in the target prompt. As an example, in the first row of Figure 5(left), GNRI exclusively converts the cat into a dog without changing the text in the sign whereas other methods either fail to generate a proper dog or change the text in the sign. Results for SD2.1 in Appendix H. For additional qualitative results see also Appendix H,G and K. **User Study.** We further evaluated editing quality using human raters. We followed the evaluation process of Mokady et al. (2023) and asked human raters to rank edits made by the three leading methods: Ours, ReNoise (Garibi et al., 2024), and ExactDPM (Hong et al., 2024). Unfortunately, the code for TurboEdit (Wu et al., 2024) was not available at the time of our evaluation. 100 images were rated, 12 images were provided by the authors of (Mokady et al., 2023) and the rest were randomly selected from COCO (Lin et al., 2014) and PieBench (Ju et al., 2023) benchmarks. Three raters for each image were recruited through Amazon Mechanical Turk and were tasked to choose the image that better applies the requested text edit while preserving most of the original image. GNRI was preferred with 71.6% preference compared to ReNoise's (Garibi et al., 2024) 15.9% and ExactDPM (Hong et al., 2024) 12.5%.

**Quantitative results.** Following (Wallace et al., 2023; Huberman et al., 2023; Garibi et al., 2024), we evaluate the results using two complementary metrics: LPIPS (Zhang et al., 2018) to quantify the extent of structure preservation (lower is better) and a CLIP-based score to quantify how well the generated images comply with the text prompt (higher is better). Metrics are averaged across 100 MS-COCO images. Figure 5(right) illustrates that editing with GNRI yields a superior CLIP and LPIPS score, demonstrating the ability to perform state-of-the-art real image editing with superior fidelity. Following (Wu et al., 2024) we also evaluated our approach on the newly introduced PieBench (Ju et al., 2023) dataset. Table 1 shows that our method more effectively follows the text guidance while preserving the background, outperforming current SoTA techniques. These evaluations further affirm the findings derived from the user study.

## 5.4 SEED INTERPOLATION AND RARE CONCEPT GENERATION

We evaluate the benefit of our inversion in a second task, of selecting latent seeds for image generation. It has been shown (Samuel et al., 2024; 2023) that text-to-image diffusion models poorly

Table 3: **Inversion Quality Impact on Rare Concept Generation:** We assess image generation using a pre-trained classifier's accuracy, comparing NAO (Samuel et al., 2023) (with DDIM inversion), ReNoise (Garibi et al., 2024), ExactDPM Zhang et al. (2023) and GNRI. We report average per-class accuracy for Head (over 1M samples), Medium, and Tail (rare classes < 10K samples). GNRI enhances rare and medium concept accuracy without sacrificing overall performance.

| | ImageNet1k in LAION2B | | | | | | |
| | **Head** | **Medium** | **Tail** | **Total Acc** | **FID** | $\hat{T}_{Init}$ | $\hat{T}_{Opt}$ |
| | n=235 | n=509 | n=256 | | | (sec) | (sec) |
| **Methods** | #>1M | 1M>#>10K | 10K># | | | | |
|---|---|---|---|---|---|---|---|
| DDIM inversion | 98.5 | 96.9 | 85.1 | 94.3 | 6.4 | 25 | 29 |
| ExactDPM (Hong et al., 2024) | 98.1 | 96.8 | 85.3 | 94.1 | 7.1 | 240 | 32 |
| ReNoise (Garibi et al., 2024) | 98.5 | 97.0 | 85.3 | 94.4 | 6.9 | 24 | 28 |
| GNRI (**ours**) | **98.6** | **97.9** | **89.1** | **95.8** | **6.3** | **17** | **25** |

generated rare concepts, and this can be improved by selecting better noise latent $z_T$. To address this issue, SeedSelect (Samuel et al., 2024) takes a few images of a rare concept as input and uses a diffusion inversion module to iteratively refine the obtained seeds. These refined seeds are then used to generate new plausible images of the rare concept. NAO (Samuel et al., 2023) extends this by introducing new paths and centroids for seed initialization. Both methods rely on DDIM Inversions, crucial for initial seed evaluation. GNRI provides an alternative for precise inversion seeds, aiming for improved image quality and semantic accuracy. It's important to highlight that most inversion techniques (Mokady et al., 2023; Wallace et al., 2023; Huberman et al., 2023; Brack et al., 2024) are not applicable in this context as they necessitate extra parameters for image reconstruction, which impedes the straightforward implementation of interpolation.

**Interpolation and centroid finding:** We evaluate GNRI by following the experimental protocol of (Samuel et al., 2023) and compare images generated by four methods: DDIM, ReNoise (Garibi et al., 2024), ExactDOM (Hong et al., 2024) and GNRI. Evaluation is conducted based on FID score and classification accuracy (by a pre-trained classifier). Results are presented in Table 2. Notably, initializing with GNRI seeds consistently results in higher-quality images both in interpolation paths and seed centroids.

**Rare concept generation:** We further show the effect of our seed inversions on the performance of NAO centroids with SeedSelect for rare concept generation. Specifically, we compared images generated by SeedSelect, initialized with NAO using DDIM inversion, ReNoise, ExactDPM, and GNRI, as shown in Table 3). We followed the evaluation protocol of (Samuel et al., 2024; 2023) on ImageNet1k classes arranged by their prevalence in the LAION2B dataset (Schuhmann et al., 2022). The results, summarized in Table 3, demonstrate that our inversion method significantly boosts performance, both in classification accuracy and FID, compared to other methods. Furthermore, our inversions yield a more precise and effective initialization point for SeedSelect (Samuel et al., 2024), resulting in notably quicker convergence without compromising performance, in all categories (head, medium, and tail) along with a high gap at tail.

# 6 SUMMARY

Image inversion in diffusion models is vital for various applications like image editing, semantic augmentation, and generating rare concept images. Current methods often sacrifice inversion quality for computational efficiency, requiring significantly more compute resources for high-quality results. This paper presents Guided Newton-Raphson Inversion (GNRI), a novel iterative approach that balances rapid convergence with superior accuracy, execution time, and memory efficiency. Using GNRI opens the door for high-quality real-time image editing due to its efficiency and speed. GNRI requires no model training, fine-tuning, prompt optimization, or additional parameters, and is compatible with all pre-trained diffusion and flow-matching models with deterministic schedulers.

ACKNOWLEDGMENTS

We would like to express our sincere gratitude to Prof. Nir Sochen for the intriguing discussion during the course of this study.

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

# Appendix

## A    NEWTON METHOD FOR MULTIVARIABLE SCALAR FUNCTION

In this section, we present the formulation of Newton's method for zero crossing of a multi-variable scalar function. For a vector $\mathbf{x} \in \mathbb{R}^D$ and a function $f(\mathbf{x}) : \mathbb{R}^D \to \mathbb{R}$ we are looking for the roots of the equation $f(\mathbf{x}) = 0$. Assuming that the function $f$ is differentiable, we can use Taylor expansion:

$$f(\mathbf{x} + \delta) = f(\mathbf{x}) + \nabla f \, \delta^T + o(||\delta||^2) \tag{A1}$$

For every $\delta, \mathbf{x} \in \mathbb{R}^D$. The idea of Newton's method in higher dimensions is very similar to the one-dimensional scalar function; Given an iterate $\mathbf{x}^k$, we define the next iterate $\mathbf{x}^{k+1}$ by linearizing the equation $f(\mathbf{x}^k) = 0$ around $\mathbf{x}^k$ as above, and solving the linearized equation $f(\mathbf{x}^k + \delta) = 0$. Dropping the higher orders we then solve the remaining linear equation for $\delta$. In fact, in the scalar case, the solution for $\delta$ is not unique, with each solution potentially determining a different direction of propagation of the iterate $\mathbf{x}^k$ in the high-dimensional input space. The optimal direction cannot be known in advance, and some have proposed selecting the propagation direction by imposing constraints, such as minimizing $\delta$ (Polyak & Tremba, 2020). We adopt a simple, non-sparse, yet effective solution, that allows the scheme flexibility in deriving the direction of propagation:

$$\delta = -\frac{1}{D} \left[ \frac{1}{\frac{\partial f}{\partial x_1}}, \frac{1}{\frac{\partial f}{\partial x_2}}, ..., \frac{1}{\frac{\partial f}{\partial x_D}} \right] f(\mathbf{x}^k) \tag{A2}$$

where $x_i$ indicates the $i$-th component of vector $\mathbf{x}$. The above can be easily proven as a solution by substituting $\delta$ into Eq. A1, dropping the higher orders setting it equal to zero. Using the relation, $\delta = \mathbf{x}^{k+1} - \mathbf{x}^k$ we get the final iterative scheme:

$$x_i^{k+1} = x_i^k - \frac{1}{D} \frac{f(\mathbf{x}^k)}{\frac{\partial f}{\partial x_i}(\mathbf{x}^k)} \tag{A3}$$

Note that this equation is component-wise i.e. each component is updated separately, facilitating the solution.

## B    DISCUSSION ON THE NEWTON-RAPHSON METHOD FOR FUNCTIONS WITHOUT ZERO CROSSINGS

Note that our objective function in Eq. 9 is a non-negative function. When the target function $\mathcal{F} \geq 0$ and does not have zero crossings,

the NR procedure may converge to a (non-zero) minima. A common example is the function $f(x) = x^2 + 1$ that has no real roots, but NR converges to its minimum. Note that while the Newton-Raphson method is typically used to find zero crossings of a function, our approach applies it to drive convergence toward local minima of the objective function (being locally convex around this point). The analysis of this type of convergence within the Newton-Raphson framework is left for future numerical analysis research.

## C    GNRI SCHEME CONTRACTION MAPPING

Fixed-point iterations are guaranteed to converge if the operator is contractive in the region of interest. Formally, a mapping $T$ is contractive on its domain if there exists $\rho < 1$ (also called Liphshitz constant) such that $\|T(x) - T(y)\| \leq \rho \|x - y\|, \forall (x, y) \in X$, where $\| \cdot \|$ denotes a norm. In the following we will demonstrate empirically that our GNRI scheme introduces contraction mapping, a property needed for convergence of the scheme. Contractive mappings ensure that each iteration moves closer to the limit, with the distance shrinking at a predictable rate. The rate of contraction (the Lipschitz constant) can be used to give bounds on how fast the error decreases, helping to assess the efficiency of the scheme. Next, we show empirically that, the mapping is contractive in our region of interest.

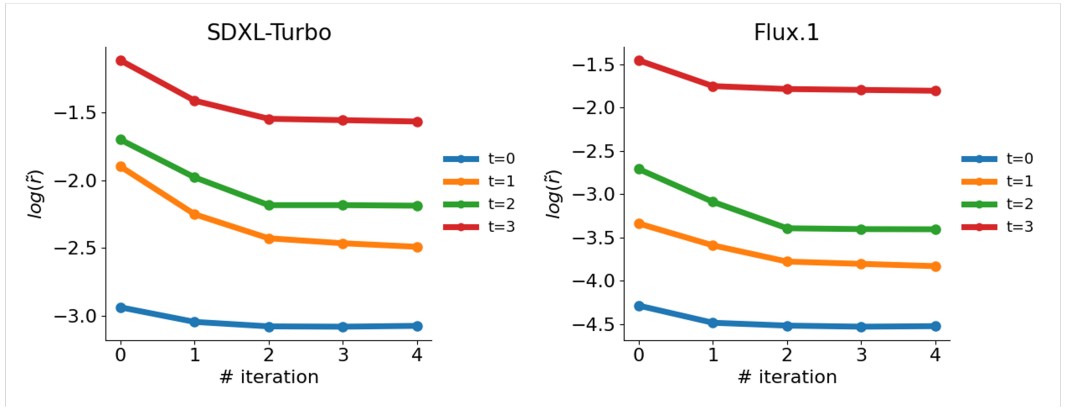

Figure C1: Contraction Mapping. Contraction mapping for two fast diffusion models are shown (in log-scale), at each time-step. The scheme converges in 2-3 iterations with a notable contraction occurring in the first iteration, assisting the rapid convergence. $t = 0$ is close to the image side and $t = 3$ close to seed.

According to Eq. 10 the GNRI scheme for the solution $\mathcal{F}(z) = 0$ (defined in Eq. 9), at each diffusion time-step, is given by:

$$z_i^{k+1} = z_i^k - \frac{1}{D} \frac{\mathcal{F}(z^k)}{g_i(z^k)} \tag{C1}$$

where $g_i := \frac{\partial \mathcal{F}}{\partial z_i}$ is the partial-derivative of $\mathcal{F}$ with respect to the variable $z_i$, while $i \in [0, 1, ..., D]$ indicates $z$'s components. We can rewrite the NR iteration as the fixed-point iteration $\tilde{f}(z) = z$ by setting $\tilde{f}(z)$ as:

$$\tilde{f}_i(z^k) := z_i^k - \frac{1}{D} \frac{\mathcal{F}(z^k)}{g_i(z^k)} \tag{C2}$$

Note that the fixed point of the newly introduced $\tilde{f}$ is the solution of $\mathcal{F}(z) = 0$.

Let us now define a *new* residual function:

$$\tilde{r}(z^k) := ||\tilde{f}(z^k) - z^k|| \tag{C3}$$

By substitution we get:

$$\tilde{r}(z^{k+1}) = ||\tilde{f}(z^{k+1}) - z^{k+1}|| = ||f(z^{k+1}) - \tilde{f}(z^k)|| \tag{C4}$$
$$\tilde{r}(z^k) = ||\tilde{f}(z^k) - z^k|| = ||z^{k+1} - z^k||$$

Therefore contraction mapping requirement for $\tilde{f}$: $||\tilde{f}(z^{k+1}) - \tilde{f}(z^k)|| \le \rho ||z^{k+1} - z^k||$ can be written as:

$$\tilde{r}(z^{k+1}) \le \rho \, \tilde{r}(z^k) \tag{C5}$$

and the contraction mapping is satisfied if $\tilde{r}$ follows an exponential decay ($r(z^k) \le \rho^k \, r(z^0)$), with some $\rho < 1$. Note that $\tilde{r}(z^k)$ given in Eq. C3 can be computed at each iteration step. In Fig. C1 we plot $\tilde{r}(z^k)$, tested on 5,000 image caption pairs from the COCO dataset (the same as used in Sec. 5.1) showing that $\tilde{r}(z^k)$ indeed decays exponentially (linearly in log-scale), till convergence. This behaviour therefore implies that GNRI iteration is a contraction mapping. Note that our contraction mapping does not converge to zero, namely the fixed point, since our objective function $\mathcal{F}$ may not have an exact solution. Computing contraction rate $\rho$ for GNRI shows $\rho \approx 0.50$ (at the first iteration) while $\rho \approx 0.90$ for FPI indicating the faster convergence of GNRI.

Note that we further show in the main paper the convergence of the equation residual $\hat{r} = ||f(z_t) - z_t||$ (EQ. 7) as an indicator for convergence toward the inversion solution.

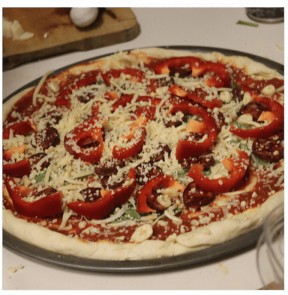 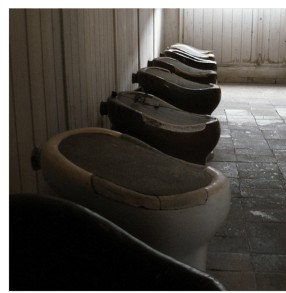 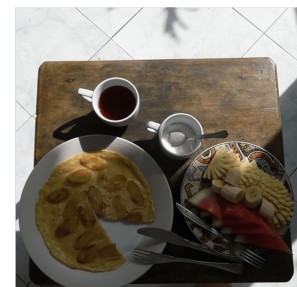

*"Your guess is as good as mine as to what these objects are."*

*"A thing is in the outline and it shows up like something"*

*"Food on a train with a pie and some vegetables."*

Figure D1: **Failure cases:** GNRI fails to converge where the prompt and image do not align.

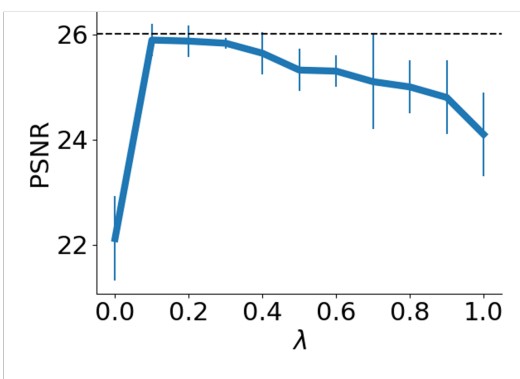

Figure D2: The influence of $\lambda$ on reconstruction performance. We observe that employing no regularization ($\lambda = 0$) leads to poor reconstruction, while $\lambda = 0.1$ typically yields the highest reconstruction accuracy. As $\lambda$ increases, reconstruction accuracy declines, possibly because assigning excessive weight to the prior undermines the root-finding objective.

## D  FAILURE CASES

Here we provide information about an analysis to find failure cases for convergence. We ran inversion and reconstruction, in scale, on COCO2017 (118k caption-image pairs) and found that in 95.4% of cases, GNRI successfully converged to a solution. Specifically, residuals went down from $\sim 1$ (for DDIM) to $< 10^{-4}$ indicating convergence, and the PSNR was $> 25.7$ indicating good solutions. The remaining 4.6% were samples with incorrect captions, see Fig. SD1. These results imply that lack of convergence may indicate text and image miss-alignment which we consider for future work.

## E  PRIOR TRADE-OFF PARAMETER

Figure D2 shows the impact of the weight parameter $\lambda$ on the reconstruction accuracy. We observe that using no regularization ($\lambda = 0$) results in poor reconstruction, whereas setting $\lambda = 0.1$ achieves the highest reconstruction accuracy, underscoring the importance of the prior in the objective function. There is a gradual decrease with larger $\lambda$ values, where at $\lambda = 1$ the PSNR starts to decline faster, likely because placing too much emphasis on the prior compromises the residual root-finding objective.

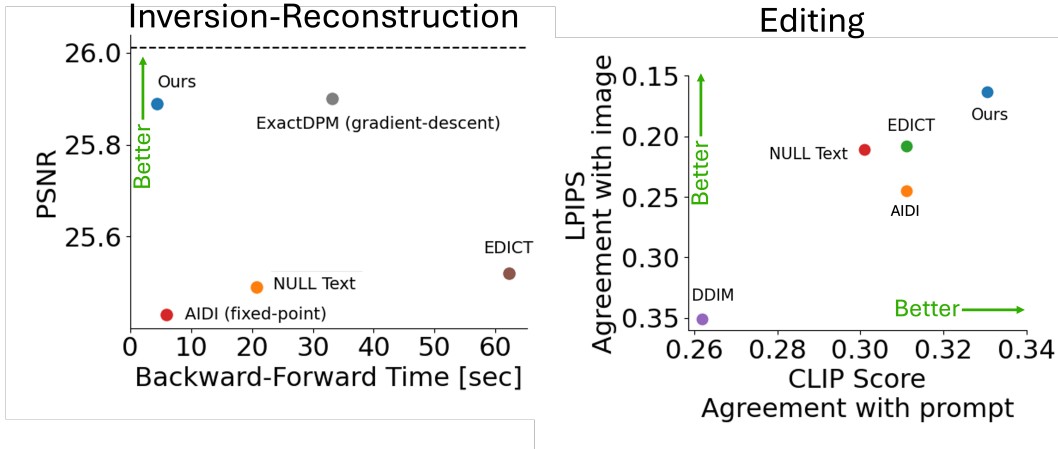

Figure E1: **(Left) Inversion Results:** Mean reconstruction quality (y-axis, PSNR) and runtime (x-axis, seconds) on the COCO2017 validation set. Our method achieves high PSNR while reducing inversion-reconstruction time by a factor of $\times 4$ to $\times 14$.**(Right) Evaluation of editing performance:** GNRI achieves superior CLIP and LPIPS scores, indicating better compliance with text prompts and higher structure preservation.

## F  SEED EXPLORATION & RARE CONCEPT GENERATION

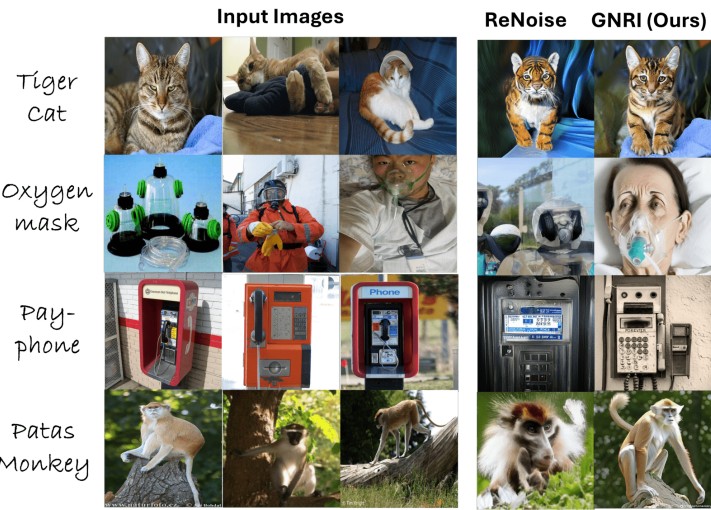

Figure F1: Generating rare concepts based on a few examples with the method of (Samuel et al., 2023) that heavily depends on the diffusion-inversion quality. In our comparison, we evaluate GNRI alongside ReNoise (refer to the discussion in the main paper). ReNoise frequently struggles to produce a realistic representation of certain objects (such as Oxygen-mask or Patas Monkey) or an accurate depiction of specific concepts (like Tiger-cat or Pay-phone). However, the use of GNRI rectifies these issues in the results. For a detailed quantitative comparison, please refer to the main paper. See the main paper for a quantitative comparison.

Figure SF1 further displays results for the rare-concept generation task introduced in (Samuel et al., 2024; 2023). The objective is to enable the diffusion model to generate concepts rarely seen in its training set by using a few images of that concept. Both methods in (Samuel et al., 2024; 2023) utilize diffusion inversion for this purpose. In the main paper, we presented experiments showcasing the impact of our new inversion process on the outcomes of (Samuel et al., 2024; 2023).

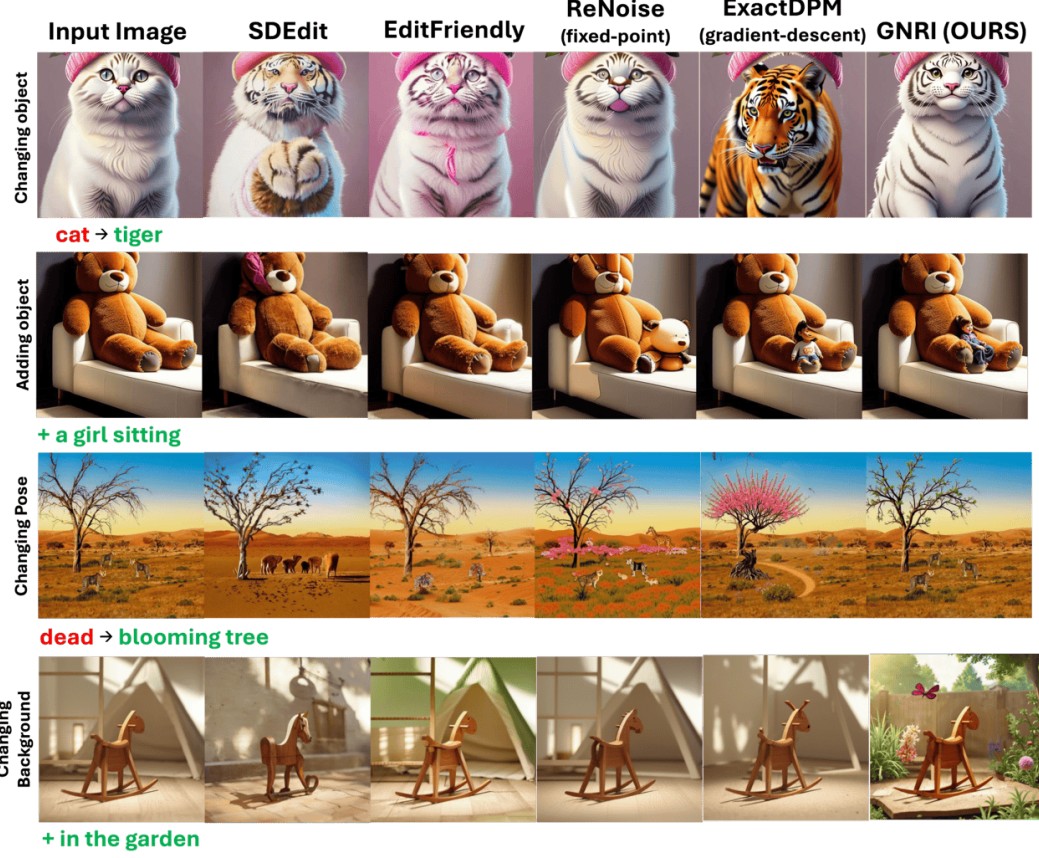

Figure F2: Qualitative comparison for **image editing** on the PIE-benchmark. Illustrates how alternative approaches may struggle to preserve the original structure or fail to edit the image, while GNRI succeeds in editing the image properly.

We provide qualitative results based on NAO (Samuel et al., 2023) centroid computation when utilizing GNRI, contrasting them with those obtained through DDIM inversions. The illustrations demonstrate that GNRI is capable of identifying high-quality centroids with correct semantic content compared to centroids found by DDIM.

## G    ADDITIONAL QUALITATIVE RESULTS ON PIE-BENCHMARK

**Additional quantitative results on the PIE-Benchmark.**    Figure F2 provides quantitative results on the PIE-Benchmark using Flux.1. It highlights how alternative methods often fail to maintain the original structure or miss key elements specified in the target prompt, whereas GNRI effectively achieves accurate and proper image editing.

**Quantitative results with Stable Diffusion 2.1.**    We present a quantitative comparison between GNRI and baseline methods, utilizing latent diffusion SD2.1 (Rombach et al., 2022). Figure E1(a) illustrates the image inversion-reconstruction performance, while Figure E1(b) highlights the editing performance, evaluated using LPIPS and CLIP-score metrics. The results demonstrate that GNRI surpasses all other inversion methods in terms of reconstruction quality, which subsequently improves editing performance.

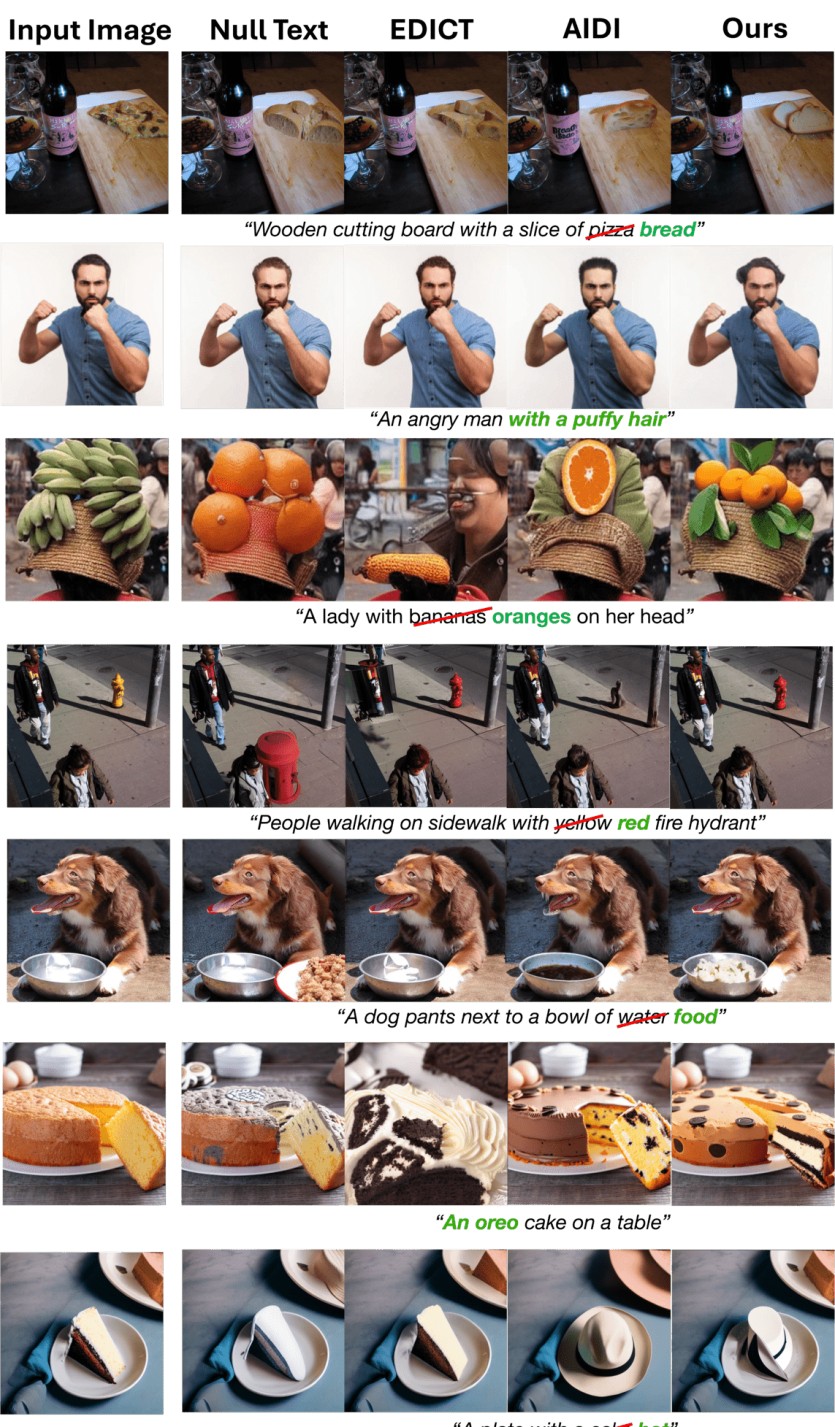

Figure F3: Qualitative comparison for **image editing:** Illustrates how alternative approaches may struggle to preserve the original structure or overlook crucial components specified in the target prompt, while GNRI succeeds in editing the image properly.

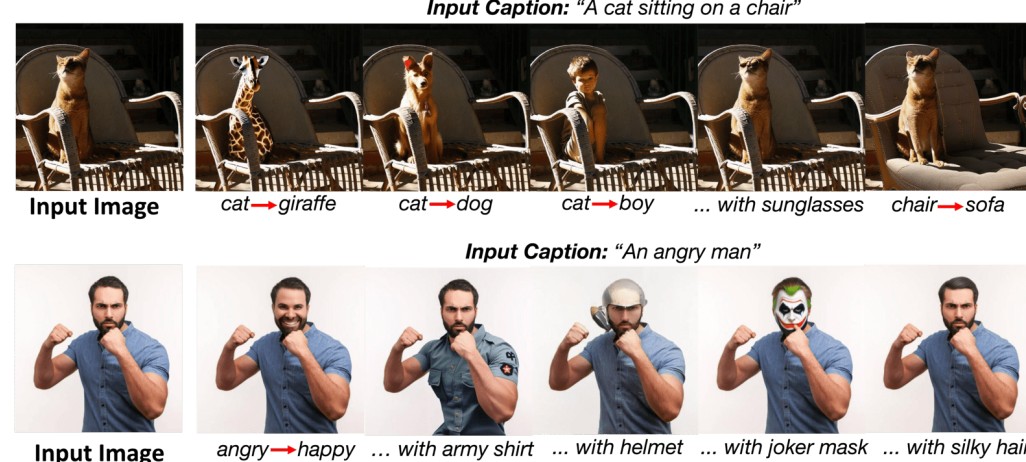

Figure F4: Various editing with same input: Note the GNRI capability in both subtle and extensive changes as one would expect from the particular prompt change.

Table H1: PSNR vs. NFE results using SDXL Turbo on the COCO validation set. Our method achieves the highest PSNR with the fewest NFEs, compared to other methods.

|  | PSNR | NFE |
|---|---|---|
| ReNoise Garibi et al. (2024) | 18.1 | 36 |
| ExactDPM Hong et al. (2024) | 20.0 | 40 |
| TurboEdit Wu et al. (2024) | 22 | 8 |
| **GNRI (Ours)** | **23.9** | **6** |

Table H2: Inversion comparison with Edit Friendly on SDXL Turbo (COCO validation set). While DDPM achieves higher PSNR through stochastic inversion, our deterministic approach (GNRI) provides competitive results with significantly faster inversion time (0.521s vs 32.3s).

|  | PSNR | Inv Time |
|---|---|---|
| EditFriendly Huberman et al. (2023) | **26.1** | 32.3 |
| **GNRI (Ours)** | 23.9 | **0.521** |

# H  ADDITIONAL QUALITATIVE AND QUANTITATIVE RESULTS - COMPARISON TO BASELINES

In this section, we present additional qualitative comparisons involving GNRI and baseline methods.

Figures F3 provide further comparisons for real-image editing, using SD2.1. These figures illustrate how alternative approaches may struggle to preserve the original structure or overlook crucial components specified in the target prompt. For example, n the first row of Figure F3, GNRI correctly transforms the pizza into bread, while other methods generate a bad looking bread. Row three shows an example of GNRI success (bananas to oranges), where other alternatives totally fail.

Figure F4 shows the outcomes of various edits on the same image, providing additional confirmation that inversion with GNRI yields modification of the pertinent parts in the image while maintaining the original structure.

Table H1 provides PSNR vs NFE (Neural Function Evaluations) results using SDXL Turbo on the COCO validation set. Our method requires 6 NFEs, calculated as 4 steps with an average of 1.5 NR iterations per step (some steps needing 1 iteration, others 2). As shown in the table below, our approach achieves the highest PSNR while using the fewest NFEs compared to other methods.

Table I1: Comparison of Anderson Acceleration and the Guidance term for out-of-distribution COCO samples. A higher likelihood is better.

| | mean likelihood $q(z_t\|z_{t-1})$ | PSNR | Inv Time |
|---|---|---|---|
| Anderson Acceleration | 0.34 | 18.3 | Inf |
| **Guidance Regularizer (Ours)** | **0.82** | **25.8** | **0.4 s** |

Table J1: Memory usage and convergence time for different methods on A100 and RTX 3090Ti.

| | A100 (VRAM 40GB) | | RTX 3090Ti (VRAM 24GB) | |
|---|---|---|---|---|
| | Memory | Inv Time | Memory | Inv Time |
| AIDI Pan et al. (2023) | 15GB | 5 sec | 16 GB | 13 sec |
| ExactDPM Hong et al. (2024) | 30GB | 17 sec | 20 GB | 25 sec |
| **GNRI (Ours)** | 28GB | 0.4 sec | 19 GB | 1 sec |

Table H2 compares inversion performance with Edit Friendly Huberman et al. (2023) using SDXL Turbo on the COCO validation set. Edit Friendly achieves the maximum PSNR possible given the VAE distortion, because it does a stochastic DDPM inversion, saving additional noise matrices at every step and "overfitting" the inverted image. As a result, it requires extensive inversion time (more than 30 seconds), whereas our GNRI reaches a relatively high PSNR in under a second. Our approach solves deterministic inversion (DDIM or Euler) making it suitable for applications like interpolation and rare concept generation, where DDPM is not feasible.

## I    COMPARISON TO ANDERSON ACCELERATION

The guiding term introduced in Sec 4.2 might appear similar to Anderson acceleration used in Pan et al. (2023), which aims to allow more gradual changes from previous estimates. However, One key difference is that our guidance term introduced in Eq.(9) does not pull $z_t$ toward $z_{t-1}$ but towards the mean $\mu_t$. By that, it directs the Newton-Raphson method toward solutions having a high likelihood, using "side-information" about the noise. In contrast, Anderson acceleration relies on previous values of the implicit function $f(z_i)$ and residuals but lacks awareness of the latent distribution (see for example, line 13 in Algorithm 1 of (Pan et al., 2023)).

In GNRI, keeping $z_t$ close to $\mu$, encourages solutions that are more "in-distribution" and aligned with the diffusion model. In other words, if $z_{t-1}$ starts far out-of-distribution, keeping $z_t$ close to $z_{t-1}$ is likely to yield worse solutions compared to guiding $z_t$ toward inputs that the model expects to receive.

We illustrate this effect in table I1. We analyzed the inversion of 300 COCO images that the model would consider to be non-typical by selecting the samples with the lowest likelihood under $q(z_0)$ (Eq.(1)). The table shows that GNRI drives the latents toward significantly higher likelihood solutions (mean $q(z_t\|z_{t-1})$), leading to substantially better reconstructions (higher PSNR) than Anderson acceleration.

More broadly, our guidance term could potentially be applied to other methods, such as fixed-point iteration approaches like those proposed by Pan et al. It would also be interesting to find methods that can include more domain knowledge into the inversion process.

## J    MEMORY USAGE

We compared the memory usage of our approach, with AIDI Pan et al. (2023) and ExactDPM (Zhang et al. (2023)) on a A100 GPU (40 GB VRAM) and a RTX 3090Ti GPU (24GB VRAM). Results are given in table J1. Our approach consumes about the same GPU memory as ExactDPM, however, it converges faster about x43 times than ExactDPM on an A100 GPU, and x25 faster on an RTX 3090 GPU.

## K    MULTI OBJECT EDITING

We conducted several qualitative experiments to evaluate the capabilities of our method in editing multiple objects simultaneously. Our results demonstrate that the method can effectively handle complex scenes, allowing for the independent manipulation of several objects without compromising the quality of individual edits. As illustrated in Figure K1, our GNRI model successfully performs multi-object transformations while maintaining scene coherence and natural composition - from converting multiple dogs to cats while preserving the scene layout, to simultaneously transforming different food items (steak to apple, eggs to pancakes) on a plate. These results underscore GNRI's robustness in handling multiple concurrent edits while maintaining high visual quality and semantic coherence across the transformed scenes.

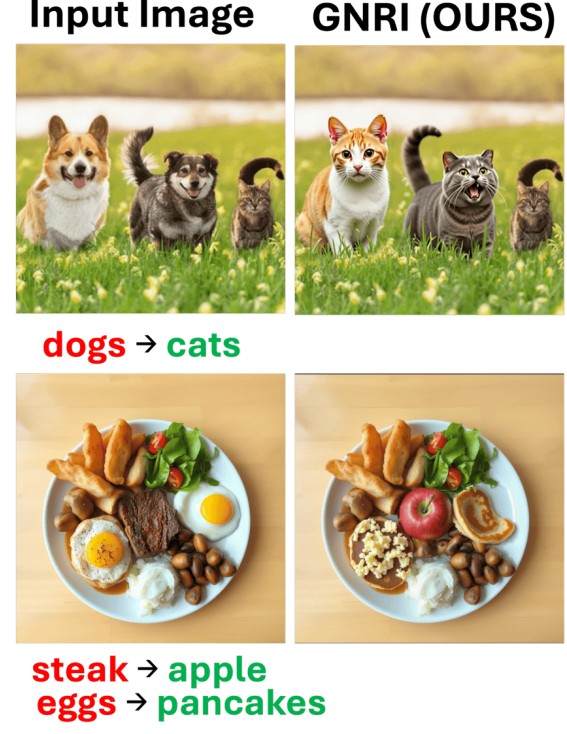

Figure K1: Multi-object editing capabilities of GNRI. Top: Simultaneous transformation of multiple dogs into cats. Bottom: Concurrent food item conversion on a plate (steak→apple, eggs→pancakes), demonstrating preservation of scene composition and context.

