# OpenReview forum: "Lightning-Fast Image Inversion and Editing for Text-to-Image Diffusion Models"
_ICLR.cc/2025/Conference — ICLR 2025 Poster_

### Official Review · Reviewer_TzFZ · 2024-10-16

**Soundness:** 3
**Presentation:** 2
**Contribution:** 3
**Rating:** 5
**Confidence:** 4

**Summary:**

This paper introduces a new inversion method for image editing in T2I DMs. It uses the Newton-Raphson (NR) method to improve the speed and accuracy of the inversion process. The authors present two main contributions: applying a norm to the NR method (Sec 4.1) and using an anchoring technique (Sec 4.2). Experimental results show that they can edit real images lightning-fast.

**Strengths:**

1. **Innovative Approach**: The use of the Newton-Raphson method helps overcome common problems found in other iterative inversion techniques (AIDI, ReNoise, ExactDPM), making the process more efficient.

2. **Comprehensive Experiments**: The paper provides experiments on three practical tasks, showing that GNRI well compared to existing methods, especially in terms of speed.

**Weaknesses:**

1. **Questionable Contribution**: Although the idea of employing NR seems innovative, the contribution in Sec 4.1 about applying the norm seems standard and not very innovative. Optimizing norms is a common practice in optimization. For example, on page 9 of ReNoise, the convergence analysis is performed using the l2-norm, and ExactDPM also applies gradient descent to the l2-norm. Actually, anyone who tries to apply Newton method would do the same thing naturally, because calculating Jacobian is impossible while calculating the gradient of the *norm* is very easy in PyTorch. Therefore, this point does not stand out.

2. **Similarity to Other Methods**: The guiding regularizer introduced in Sec 4.2 appears similar to Pan et al.'s Anderson acceleration, which also aims to prevent large changes from previous estimates. This similarity raises questions about the originality of the approach. On the contrary, the significant drop in performance when $\lambda=0$ in Figure C2 can be seen as an indication of the instability of NRI.

3. **Memory Usage**: The Newton-Raphson method needs gradients, which can lead to high memory use. (It must be using torch.grad, I guess.) Other methods, like AIDI and the forward step method in ExactDPM, do not rely on gradients and may be more efficient, in terms of memory usage. ExactDPM actually used a smaller GPU than A100.

4. **Some writings are confusing**: The description of $\mathcal{F}$ is confusing because it is defined in two different ways: as a function from $\mathbb{R}^D \to \mathbb{R}^D$ in L203 and again as a function from $\mathbb{R}^D \to \mathbb{R}^+$ in Eq (9). This inconsistency can mislead readers. Also, in line 577, the citation for Hong et al. is duplicated. In line 637, it should be noted that EDICT was presented at CVPR *2023*, not 2022.

5. **Lack of Results for Rare Concept Generation**: The paper does not provide strong quantitative results for generating rare concepts. For example, Table 3 does not show significant improvements over existing methods.

**Questions:**

1. In L238, should it reference Eq. 4 instead of Eq. 5?
2. Could you provide more results for the rare concept generation?

---

> ### Comment · Reviewer_TzFZ · 2024-11-13
> **I add the questions related to the weaknesses.**
>
> As the weaknesses were not addressed further in the questions, I have organized additional questions to help strengthen the manuscript. All the additional questions are related to the weaknesses I have written.
>
> 3. **Norm Application**: Could you explain more clearly how your idea of applying norms in the Newton-Raphson approach for diffusion model inversion is different from or improves on existing methods, in terms of novelty? This would help to appeal the novelty.
>
> 4. **Comparison with Anderson Acceleration**: Could you provide a detailed comparison between your guiding regularizer and Anderson acceleration (Pan et al., 2023), focusing on any key differences or improvements. Also, any extra analysis on the instability at $\lambda=0$ would be useful.
>
> 5. **Memory Usage Analysis**: Could you add a comparison of the memory usage of your method versus other approaches (Pan et al., 2023; Hong et al., 2024), with some examples across different GPU sizes?
>
> If these points are addressed, I would consider adjusting the score. Thank you very much for your patience.

---

> ### Author Response · Authors · 2024-11-20
> **Thank you for the review! (response part 1/2)**
>
> Thank you for your detailed and insightful feedback. We address all your concerns below.
>
> **1. Applying the norm seems standard.**
> We thank the reviewer for the comment. We do not claim that the novelty of the paper lies in applying the norm of the residual (Eq (7)), but in the following: (1) the novel application of Newton-Raphson for diffusion inversion and (2) the prior added to the objective based on the known distribution $q(z_t|z_{t-1})$ to guide optimization toward in-distribution solutions.
> Following the comment, we revised the text to clarify applying the norm has been done in previous work, and included the references provided by the reviewer.
>
> **2. The guiding regularizer … appears similar to Anderson acceleration, which also aims to prevent large changes from previous estimates.**
> Thank you for this comment. One key difference is that the guidance term introduced in Eq.(9) does not pull $z_t$ toward $z_{t-1}$ but towards the mean $\mu_t$. By that, it directs the Newton-Raphson method toward solutions having a high likelihood, using “side-information” about the noise. In contrast, Anderson acceleration relies on previous values of the implicit function $f(z_i)$ and residuals but lacks awareness of the latent distribution (see for example, line 13 in Algorithm 1 of Pan et al.).
>
> In GNRI, keeping $z_t$ close to $\mu$, encourages solutions that are more “in-distribution” and aligned with the diffusion model. In other words, if $z_{t-1}$ starts far out-of-distribution, keeping $z_t$ close to $z_{t-1}$ is likely to yield worse solutions compared to guiding $z_t$ toward inputs that the model expects to receive.
> We illustrate this effect in the table below. We analyzed the inversion of 300 COCO images that the model would consider to be non-typical by selecting the samples with the lowest likelihood under $q(z_0)$ (Eq.(1)). The table shows that GNRI drives the latents toward significantly higher likelihood solutions (mean $q(z_t|z_{t-1})$), leading to substantially better reconstructions (higher PSNR) than Anderson acceleration.
> More broadly, our guidance term could potentially be applied to other methods, such as fixed-point iteration approaches like those proposed by Pan et al. It would also be interesting to find methods that can include more domain knowledge into the inversion process.
> | Flux.1-schnell         | Out-of-Distribution COCO            |                            |                 |
> |-------------------------|--------------------------------------|----------------------------|-----------------|
> |                         | **mean likelihood q(z_t\|z_t-1)** (**Higher is better**)       | **PSNR**                       | **Inv Time**        |
> | Anderson Acceleration   | 0.34                               | 18.3                       | Failed          |
> | **Guidance Regularizer (Ours)**    | **0.82**                           | **25.8**                   | **0.4 s**       |
>
> **3. There is a significant drop in performance when λ=0. Add analysis.**
> Thank you for pointing out this topic and allowing us to make it clearer. Setting $λ=0$ corresponds to the Newton-Raphson method without the guidance term (Section 7). In this case, we observed rapid convergence to solutions that lead to poor image reconstructions (Figure C2). We presume this is because the Newton-Raphson process selects solutions that solve the implicit equation, but are outside the distribution of latents observed when training the model. We provided analysis and details in Section 5.1 and in Appendix D. We’re happy to provide any specific analysis the review may still ask for.
>
> **4. Add a comparison of the memory usage of your method versus other approaches.**
> Following the reviewer’s comment we compared the memory usage of our approach, with  AIDI [1] and ExactDPM [2] on an A100 GPU (40 GB VRAM) and an RTX 3090Ti  GPU (24GB VRAM). Results are given in the table below. Our approach consumes about the same GPU memory as ExactDPM, however, it converges faster about x43 times than ExactDPM on an A100 GPU, and x25 faster on an RTX 3090 GPU.
>
> |                         | A100 (VRAM 40GB)       |                     | RTX 3090Ti (VRAM 24GB) |                     |
> |-------------------------|------------------------|---------------------|------------------------|---------------------|
> |                         | Memory                | Inv Time            | Memory                | Inv Time            |
> | AIDI                    | 15GB                  | 5 sec               | 16 GB                 | 13 sec              |
> | ExactDPM                | 30GB                  | 17 sec              | 20 GB                 | 25 sec              |
> | **GNRI (Ours)**         | **28GB**              | **0.4 sec**         | **19 GB**             | **1 sec**           |
>
> [1] Pan et al. (2023) “Effective Real Image Editing with Accelerated Iterative Diffusion Inversion”.
> [2] Hong et al. (2024) “On Exact Inversion of DPM-Solvers”.

---

> ### Author Response · Authors · 2024-11-20
> **Thank you for the review! (response part 2/2)**
>
> **5. Typos and duplicate citations.**
> Thank you for helping us improve this paper. We fixed this in the revised manuscript.
>
> **6. There is a slight improvement over existing methods on rare concept generation.**
> Thank you for your comment. The goal of rare concept generation is to enhance tail performance without compromising performance on the head categories. The results indeed show this trend where our approach demonstrates a significant absolute 3.8% gain in the tail classes (from 85.3% to 89.1%) without hurting head class accuracy. Additionally, our method achieves a substantial reduction in generation time, 10 seconds faster (a 20% improvement) compared to the best baseline. This balance of improved performance in tail classes and enhanced efficiency highlights the strength of our approach.
>
> **7. More results for rare concept generation.**
> Thank you. Further results on rare concept generation can be found in Table 2 and Appendix E.

---

> > ### Comment · Reviewer_TzFZ · 2024-11-22
> > **Thank you for your detailed response!**
> >
> > ## 1 & 2
> > Through your response, I now acknowledge that this paper demonstrates notable novelty. (1) The novel application of Newton-Raphson for diffusion inversion is indeed innovative, as I mentioned in my initial review. Regarding (2) The prior added to the objective based on the known distribution $q(z_t|z_{t-1})$ to guide optimization toward in-distribution solutions, I initially assumed that similar ideas might already exist. However, I could not find prior works explicitly employing this approach. Additionally, the comparison to Anderson acceleration further supports the originality of (2).
> >
> > ## 3
> > Thank you for the explanation. However, I feel that simply referring to the idea in (2) as a 'prior' remains insufficient to clarify its connection to existing optimization techniques, particularly given the experimental results that underscore its significance. I am curious about the origins of (2) and the reasoning behind its effectiveness. Considering that optimization theory is a well-established field, related works likely exist that could provide valuable insights into this phenomenon.
> >
> > One potential direction could involve contrasting Maximum Likelihood (ML) with Maximum A Posteriori (MAP). For example, linking $\lambda \neq 0$ to MAP and $\lambda = 0$ to ML might offer a clearer explanation. Additionally, (2) appears somewhat similar to the anchor-based acceleration technique discussed in [A].
> >
> > Further theoretical investigation would undoubtedly strengthen the paper.
> >
> > ## 4, 5, 6 & 7
> > Thank you for addressing my concerns. Based on your response, I would like to adjust my score to 5.
> >
> > [A] D. Kim, Accelerated proximal point method for maximally monotone operators. Mathematical Programming, 190(1–2):57–87, 2021.

---

> > > ### Author Response · Authors · 2024-11-24
> > >
> > > Thank you for highlighting the novelty and originality of this work. Also, thank you for suggesting the analogy to ML-vs-MAP. We will add that connection to provide further insight as you suggested. Thank you!

---

### Official Review · Reviewer_xAUW · 2024-10-28

**Soundness:** 4
**Presentation:** 3
**Contribution:** 4
**Rating:** 8
**Confidence:** 4

**Summary:**

This paper introduces an efficient method for accelerating image inversion in diffusion models by refining the Newton-Raphson (NR) approach. While a direct application of NR is computationally intense due to the need for Jacobian computation, the authors optimize it with dimensionality reduction using the L1 norm and incorporate a regularization term to ensure the noise latent conforms to the original distribution of the diffusion model. Experiments on models like SDXL-Turbo and Flux showcase both the speed and quality of inversion, and additional tests demonstrate the method’s capabilities in image editing, interpolation, and generating rare objects.

**Strengths:**

1. The paper introduces an optimized approach to inversion in diffusion models, significantly accelerating the process. With inversion times as low as 0.4 seconds on an A100 GPU, this method is fast enough to enable interactive applications, a notable advancement over existing techniques.
2. By incorporating an L1 norm for dimensionality reduction and a regularization term to ensure conformity with the diffusion model's distribution, the method maintains high reconstruction quality, avoiding common issues like out-of-distribution errors in the generated latents.

**Weaknesses:**

1. Missing comparison with DDPM Inversion (An Edit Friendly DDPM Noise Space: Inversion and Manipulations) for the inversion performance in both quality and speed. Also, please include the DDPM-Inv result on PIE-Bench for image editing.
2. More visualization comparison of the proposed method against other methods for editing tasks. Includes more editing results from the PIE-Benchmark.

**Questions:**

I suggest the authors to include the experiments for the DDPM-Inv for both reconstruction and editing. This helps to understand the difference between using one trajectory vs sample multiple trajectory and record the noise(difference) and help us to determine which method is better.
Also please include more visualization results for the editing tasks. This helps us to understand the difference of different methods visually.

---

> ### Author Response · Authors · 2024-11-20
> **Thank you for the review!**
>
> Thank you for finding our approach efficient with a notable advancement over existing techniques. We address your comments below.
>
> **1. Missing comparison with DDPM Inversion for the inversion performance.**
> We thank the reviewer for the comment. The table below compares inversion performance with DDPM inversion (Edit Friendly [1]) using SDXL Turbo on the COCO validation set. Edit Friendly achieves the maximum PSNR possible given the VAE distortion, because it does a stochastic DDPM inversion, saving additional noise matrices at every step and ``overfitting” the inverted image.  As a result, it requires extensive inversion time (more than 30 seconds), whereas our GNRI reaches a relatively high PSNR in under a second. Our approach solves deterministic inversion (DDIM or Euler) making it suitable for applications like interpolation and rare concept generation, where DDPM is not feasible.
> |                       | **PSNR** | **Inversion Time** |
> |-----------------------|----------|--------------------|
> | Edit Friendly [1] | **26.1** | 32.3               |
> | **GNRI (Ours)**       | 23.9 | **0.521**          |
>
> **2. Please include the DDPM-Inv result on PIE-Bench for image editing.**
> The table below compares our approach with EditFriendly [1] on the Pie-bench benchmark. Our method outperforms EditFriendly across all metrics, demonstrating superior editing performance and faster inversion and editing. We added these comparisons to the revised version.
> | **PIE-Bench (SDXL-turbo)** | **Background Preservation**                          |               | **Clip Similarity**       |                | **Efficiency** |         |
> |----------------------------|-----------------------------------------------------|---------------|---------------------------|----------------|----------------|---------|
> |                            | **PSNR**        | **LPIPS 10^3** | **MSE 10^4**  | **SSIM 10^2** | **Whole Edited** | **Edited** | **Inverse** | **Forward** |
> | DDIM Inversion         | 18.59           | 177.96         | 184.69        | 66.86         | 23.62              | 21.20      | 0.454       | 0.445       |
> | DDPM Inversion         | 24.55           | 91.88          | 95.58         | 81.57         | 23.97              | 21.03      | 32.3        | 23.1        |
> | **GNRI (ours)**            | **30.22**           | **40.15**          | **20.10**         | **98.66**         | **26.15**              | **23.26**      | **0.521**       | **0.445**       |
>
> **3. Includes more editing results from the PIE-Benchmark.**
> We provide an additional qualitative comparison of the PIE benchmark in Figure E2 (Appendix F).
>
> [1] Huberman-Spiegelglas et al. (2024) “An Edit Friendly DDPM Noise Space: Inversion and Manipulations”.

---

> ### Comment · Reviewer_xAUW · 2024-11-20
> **Thank you for the results**
>
> Thank all the authors for addressing my concerns. I will raise my score 6->8.

---

### Official Review · Reviewer_Rr79 · 2024-10-30

**Soundness:** 3
**Presentation:** 4
**Contribution:** 3
**Rating:** 8
**Confidence:** 4

**Summary:**

To improve the accuracy of DDIM inversion approximations, previous studies, such as ReNoise, have utilized fixed-point iteration. This paper introduces a specialized fixed-point iteration known as Newton-Raphson (NR), which enables faster convergence. Because a straightforward application of the NR method is infeasible for high-dimensional points, this paper reduces dimensionality by applying the L1​-norm to the fixed-point function, transforming the Jacobian into a vector. Additionally, to prevent convergence to an off-manifold fixed point, a proximal term is added to the fixed-point function, steering the solution $z_t$ toward the mean of the forward distribution $q_t(z_t|z_{t-1})$ This complete framework is termed Guided Newton-Raphson Inversion (GNRI).

**Strengths:**

- The motivation and proposed method is clearly stated.
- As the NR provides faster convergence than naive fixed-point iteration method, the proposed method achieves high-efficiency compared with SOTA inversion methods.
- Combined with SDXL-turbo and Flux, it enables real-time image editing.
- Provided experimental results well demonstrate the effectiveness of the proposed method.

**Weaknesses:**

- The reviewer does not find significant weaknesses
- Minor: Linebreak in Figure 2.

**Questions:**

- Equation (10) requires one forward pass and one back-propagation through diffusion model (UNet). In fact, with the same number of fixed-point iterations, this requires additional cost for back-propagation compared to ReNoise. However, in Figure 4 and Table 1, the reported efficiency of the GNRI is almost same as the original DDIM and much better than ReNoise. Is there some points that the reviewer is missing?
- Efficiency analysis (Figure 4):  Could author provide PSNR vs NFE? It would be helpful to further understand the efficiency of inverse methods.
- Is this method also capable of multiple-object editing?

---

> ### Author Response · Authors · 2024-11-20
> **Thank you for the review!**
>
> Thank you for finding our approach effective, efficient and clearly stated. We kindly address your comments below.
>
> **1. How does GNRI match DDIM's efficiency despite the added back-propagation in Equation (10)?**
> DDIM requires a single UNet pass taking 0.2 seconds per step in Flux.1-Schnell, while GNRI performs 1–2 iterations (≈1.5 on average), taking 0.4 seconds, nearly double DDIM’s time. Fig. 4 visually marginalized this difference due to the axis scale, influenced by slower methods like ExactDPM (15+ seconds). In addition, backpropagation is fast thanks to PyTorch's optimized derivative computation. We clarified in the text that GNRI is 2x slower than DDIM.
>
> **2. Could authors provide PSNR vs NFE?**
> In response to the reviewer’s comment, we provide PSNR vs. NFE results using SDXL Turbo on the COCO validation set. Our method requires 6 NFEs, calculated as 4 steps with an average of 1.5 NR iterations per step (some steps needing 1 iteration, others 2). As shown in the table below, our approach achieves the highest PSNR while using the fewest NFEs compared to other methods.
> |                         | PSNR      | NFE  |
> |-------------------------|-----------|------|
> | ReNoise [1]            | 18.1      | 36   |
> | ExactDPM [2]           | 20.0      | 40   |
> | TurboEdit Wu et al [3] | 22        | 8    |
> | **GNRI (Ours)**        | **23.9**  | **6**|
>
> **3. Editing multiple objects.**
> Thank you. While this is not our main focus, we conducted several experiments that show that our method is capable of editing multiple objects simultaneously. Quantitative examples can be found in Appendix J.
>
> **4. Linebreak in Figure 2.**
> Thank you. We have fixed this in the updated PDF.
>
> [1] Garibi et al. (2024) “ReNoise: Real Image Inversion Through Iterative Noising”.
> [2] Hong et al. (2024) “On Exact Inversion of DPM-Solvers”.
> [3]  Wu et al. (2024) "TurboEdit: Instant text-based image editing"

---

> > ### Comment · Reviewer_Rr79 · 2024-11-21
> > **Thanks for the response.**
> >
> > After reviewing the comments from other reviewers and the authors' responses, I've decided to maintain a score of 8.

---

### Official Review · Reviewer_8MQs · 2024-11-03

**Soundness:** 3
**Presentation:** 3
**Contribution:** 3
**Rating:** 6
**Confidence:** 3

**Summary:**

This paper addresses diffusion inversion for fast and accurate inversion within text-to-image diffusion models. The authors propose Guided Newton-Raphson Inversion (GNRI), which leverages the Newton-Raphson (NR) numerical root-finding technique to achieve efficient and high-quality inversion. By constructing a modified DDIM inversion function as the target for NR and adding a guiding term, GNRI ensures convergence within the distribution of the latent space, overcoming common challenges in achieving accurate reconstructions. This approach supports tasks like real-time image editing, image interpolation, and rare concept generation, achieving significant speedups. The authors show that GNRI could invert images within 0.4 seconds on an A100 GPU while maintaining high fidelity, compared with popular diffusion models.

**Strengths:**

Originality: This paper introduces a novel approach to the diffusion inversion problem by leveraging the Newton-Raphson (NR) root-finding method with two modifications: (1) dimensionality reduction via a scalar formulation of the Jacobian and (2) a guiding regularization term. By reframing inversion as a fixed-point iteration (FPI) problem with an $L_1$ norm relaxation, this work combines numerical optimization techniques with diffusion models, addressing limitations in accuracy and speed seen in prior methods. The addition of the regularization term prevents out-of-distribution solutions.

Quality: The methodological foundation is rigorous, with well-defined derivations and justifications for each modification to the NR method. The paper also provides extensive empirical validation across multiple tasks, including image reconstruction, editing, interpolation, and rare concept generation. The inversion time to 0.4 seconds on an A100 GPU is impressive.

Clarity: The paper is generally clearly written and accessible.

Significance: GNRI addresses the problem of achieving efficient and accurate image inversion in text-to-image diffusion models, which is central to tasks like real-time image editing and rare concept generation. The experimental results across models (e.g., Stable Diffusion, SDXL-Turbo, Flux) demonstrate GNRI’s versatility, with the potential for broad impact in real-time generative model applications.

**Weaknesses:**

The concept of utilizing FPI methods in diffusion inversion is not completely new (like AIDI, ReNoise).

Figure 3 compares the performance of the method with an FPI baseline, which is not specified as the author mentioned two baseline FPI methods (AIDI, ReNoise).

There is a notational inconsistency in referencing Eq. 5 instead of Eq. 4 when describing the fixed-point function $f(z)$, which could lead to confusion regarding the intended inversion function. This typo gives the impression that the method still relies on the DDIM-inversion approximation. Please clarify these notations.

**Questions:**

The objective function $F(Z_t)$ in Eq. 9, which combines the residual term $r(z_t)$ and the regularization $G(z_t)$ raises questions about its role in achieving exact root-finding. The two terms in Eq. 9 seem to contradict each other, because a solution to $r(z_t) = 0$ will yield a $G(z_t) > 0$, which further leads to $F(z_t) > 0$ if $r(z_t) = 0$. This suggests that the method solves an approximation of the original inversion problem rather than the exact roots.

In Fig. 3(a), it is observed that GNRI achieves a lower residual than the non-guided NR inversion (NRI), which is unexpected since NRI has only one objective $r(z_t) = 0$. Why does GNRI, despite its additional regularization term, achieve a better residual than NRI? What is the residual level if you set $z_t = mu_t (G(z_t) = 0)$?

---

> ### Author Response · Authors · 2024-11-20
> **Thank you for the review!**
>
> Thank you for your feedback. We are encouraged that you find our approach novel, the methodological foundation to be rigorous and well-defined, with the potential for broad impact. We address your comments below.
>
>
> **1. The concept of FPI is not completely new.**
> Indeed. The reviewer is absolutely correct that solving the implicit inversion function using FPI approaches has been explored (AIDI [1], ReNoise [2]). We describe and compare these approaches in the paper. This paper is the first to introduce Newton-Raphson's (NR) solution to this problem and add a guidance term that drives solutions toward in-distribution latents. The experiments show that our approach offers significant advantages in terms of both speed and accuracy over these methods.
>
> **2. Which FPI baseline is used in Figure 3?**
> We apologize for the omission, it was plain fixed-point iterations, which is equivalent to AIDI [1] without Anderson Acceleration. We clarified this in the revised PDF.
>
> **3. Notational inconsistency in referencing Eq. 5 instead of Eq. 4.**
> Thank you for catching this!  We fixed it in the revised PDF.
>
> **4. The method solves an approximation of the original inversion problem rather than the exact roots.**
> We thank the reviewer for this comment. There is no guarantee that $F(z_t)=0$ has an exact root.  Since our objective $\mathcal{F}$ is non-negative, the Newton-Raphson (NR) method effectively tends to minimize the residual, for $\mathcal{F}$, leading to an approximate solution for Eq. (9). In practice, we observed that our Guided Newton Raphson converges to a solution where both $\hat{r}$ in Eq. (7) and $G$ in Eq. (8) reach low values, indicating that these constraints (residual and likelihood) are at the end best satisfied.
>
> **5. Why does GNRI, despite its additional regularization term, achieve a better residual than NRI?**
> Thank you for an insightful observation. Indeed, GNRI reaches a lower residual than NRI despite the addition of the guidance term. Our main hypothesis is that since $F(z)$ has many local minima (likely approximate solutions), running NRI may select a far-from-optimal minimum. Adding the guidance term, which is very smooth, may help optimization escape those local minima. This effect may become stronger when the diffusion model receives z's that are outside its training distribution, since the landscape there may be far less smooth. With the guidance, z is driven to a different part of the space and a distinct minimum (solution) with significantly lower residual values.
>
> **6. What is the residual level if you set zt=mu_t(G(zt)=0)?**
> Setting  $z_t=\mu_t(G(zt)=0)$ results in a mean residual of $5 * 10^{-4}$ on COCO val set, with a PSNR of $21.2$. Compare this with applying 2 iterations of GNRI, which results in a residual of $4.2*10^{-5}$ and PSNR of $25.6$. This demonstrates that while $\mu_t$ perfectly satisfies the guidance term, it does not necessarily provide an optimal solution for the residual term and thus results in poor reconstructions. GNRI, however, effectively reaches low values for both the residual and guidance terms, resulting in accurate inversions and high-quality reconstructions.
>
> [1] Pan et al. (2023) “Effective Real Image Editing with Accelerated Iterative Diffusion Inversion”.
> [2] Garibi et al. (2024) “ReNoise: Real Image Inversion Through Iterative Noising”.

---

### Author Response · Authors · 2024-11-20
**Author Rebuttal by Authors**

Dear Reviewers and ACs,
We were happy to see that reviewers found our approach **“novel”**, **“innovative”** (**8MQs**, **TzFZ**), **“well motivated”** (**8MQs**, **Rr79**), **“efficient”** and **“effective”** (**xAUW**, **Rr79**). They recognized our method as **“rigorous”** and with **“broad impact in real-time applications”** (**8MQs**), noting it as a **“notable advancement over existing techniques”** (**xAUW**). They found our experiments provide **“extensive empirical validation across multiple tasks”** (**8MQs**) and **“comprehensive”** (**TzFZ**). Finally, they acknowledged the results to be **“impressive”** (**8MQs**) that **“well demonstrate the effectiveness of our approach”**.
We have addressed the reviewers' concerns in our rebuttal and are open to further discussion.  Your input has been instrumental in improving our paper. Thank you!

---

### Meta-Review · Area_Chair_jeSQ · 2024-12-23

**Metareview:**

This paper proposed an image inverse and editing method based on the Newton-Raphson (NR) algorithm to find root of the implicit equation of inversion for diffusion model.  It designed a fixed-point iteration approach enabling faster convergence, guided towards the in-distribution roots. The proposed method was applied to image editing using Stable Diffusion, SDXL-Turbo and Flux with different deterministic schedulers. The approach enables fast image inversion (0.4 sec, using one A100 GPU), and was applied to image reconstruction, editing, interpolation and rare concept generation, demonstrating the efficiency and effectiveness. The reviewers mostly feel positive on the novelty and effectiveness, while the Reviewer TzFZ raised major concerns on the contribution in norm-based optimization and difference to Pan et al.'s Anderson acceleration. In the response, the authors clarified the novelty of this approach and discussed difference to Anderson acceleration with experimental results and the reviewer confirmed the novelty of this paper.  Consider the overall contribution and effectiveness recognized by the reviewers, this paper can be accepted, with the conditions that the novelty and relation to related works discussed in the post-rebuttal phases should be included into the final version.

**Additional Comments On Reviewer Discussion:**

Reviewer 8MQs rated score of 6, with positive comments on the novelty, quality, experiments, and questioned on the novelty compared to similar works, e.g.,  AIDI, ReNoise, and the objective function design. These questions were responded in the rebuttal, but the reviewer did not further respond to these responses.  Reviewer Rr79 raised questions on the required cost compared with the ReNoise, efficiency analysis and multi-object editing. After rebuttal, the reviewer retain the score of 8 in the final decision. Reviewer xAUW questioned on the missing comparison with DDPM inversion, more visualization results, and raised the score from 6 to 8 in the final decision. Reviewer TzFZ pointed out more concerns on the novelty, comparison with Anderson acceleration, memory usage, etc. These concerns were mostly addressed in the discussion phase, and the reviewer acknowledged the novelty but raised suggestion on the deeper investigation of the relation/comparison to the related literature of optimization, which is valuable for the future work.

---

### Decision · Program_Chairs · 2025-01-22

Accept (Poster)